# Neutrophils and the Systemic Inflammatory Response Syndrome (SIRS)

**DOI:** 10.3390/ijms241713469

**Published:** 2023-08-30

**Authors:** Janusz P. Sikora, Jakub Karawani, Jarosław Sobczak

**Affiliations:** 1Department of Paediatric Emergency Medicine, 2nd Chair of Paediatrics, Central Clinical Hospital, Medical University of Łódź, ul. Sporna 36/50, 91-738 Łódź, Poland; jaroslaw.sobczak@umed.lodz.pl; 2Faculty of Medicine, Lazarski University, ul. Świeradowska 43, 02-662 Warsaw, Poland; kubakarawani@gmail.com; 3Department of Management and Logistics in Healthcare, Medical University of Łódź, ul. Lindleya 6, 90-131 Łódź, Poland

**Keywords:** neutrophils, SIRS, cytokines, ROS, biomarkers, immunomodulation, antioxidants

## Abstract

We are not entirely able to understand, assess, and modulate the functioning of the immune system in clinical situations that lead to a systemic inflammatory response. In the search for diagnostic and treatment strategies (which are still far from perfect), it became very important to study the pathogenesis and participation of endogenous inflammation mediators. This study attempts to more precisely establish the role of neutrophils in individual phenomena occurring during an inflammatory and anti-inflammatory reaction, taking into account their cidal, immunoregulatory, and reparative abilities. Pro- and anticoagulatory properties of endothelium in systemic inflammatory response syndrome (SIRS) are emphasised, along with the resulting clinical implications (the application of immunotherapy using mesenchymal stem/stromal cells (MSCs) or IL-6 antagonists in sepsis and COVID-19 treatment, among others). Special attention is paid to reactive oxygen species (ROS), produced by neutrophils activated during “respiratory burst” in the course of SIRS; the protective and pathogenic role of these endogenous mediators is highlighted. Moreover, clinically useful biomarkers of SIRS (neutrophil extracellular traps, cell-free DNA, DAMP, TREMs, NGAL, miRNA, selected cytokines, ROS, and recognised markers of endothelial damage from the group of adhesins by means of immunohistochemical techniques) related to the neutrophils are presented, and their role in the diagnosing and forecasting of sepsis, burn disease, and COVID-19 is emphasised. Finally, examples of immunomodulation of sepsis and antioxidative thermal injury therapy are presented.

## 1. Introduction

Maintaining a balance between activation and suppression of immune response is one of the most important clinical objectives of researchers and physicians in the field of clinical immunology worldwide. The body’s immune response is activated by an invasion of the organism by microbes and by all types of injuries (e.g., burns or mechanical trauma, surgical procedures). The complex system of this response includes general systemic reactions, among which we can distinguish the innate (natural) response and acquired (adaptive) response, with antigen-presenting cells being the common link between the aforementioned mechanisms. The condition for eliciting an immune system response to the activity of microorganisms is breaking the continuity of skin or mucous membranes; after an injury, the protective barriers of the organism are broken, with subsequent destruction of tissues and cells, which in turn start processes intended to restore homeostasis, stimulating the regeneration of destroyed tissues and preventing infection. Small injuries result in a local defensive reaction of the body, whereas widespread injuries lead to a response with a systemic character. It should be kept in mind that this response is comprehensive, and a neurohormonal response, related, among others, to the secretion of corticotropin, cortisol, or catecholamines is activated in addition to the activation of the immune system [1,2].

The host’s inflammatory reaction must be precisely regulated and develop at an appropriate moment in order to rapidly and effectively remove the cause of the inflammation, and then to activate reconstructive processes. A key role in a response to infection or injury and in the development of inflammatory reaction is played by endothelium and by elements of nonspecific response, with cytokines acting as the signalling substances which trigger the entire inflammation cascade. As a result of their activity, endothelium cells start to produce substances which dilate the lumen of the vessels, proteins which allow leukocytes to penetrate their walls and tissue factor which activates the clotting pathway. These compounds cause fever, synthesis of acute phase proteins, and activate successive leukocytes and act as chemokines, which cause the migration of cells to the sites of inflammation. In conditions of controlled inflammatory reaction, proinflammatory mediators induce the creation of anti-inflammatory cytokines, the synthesis of which is an expression of the homeostasis of the organism, and the symptoms of inflammation gradually subside. However, if the inflammatory factor is strong enough, and if the body is weakened enough to make adaptive anti-inflammatory mechanism insufficient, then the inflammatory process becomes generalised. This results in the penetration of proinflammatory mediators into systemic circulation, and the results of their action on the cells of tissues far away from the original inflammatory process become truly destructive. Clinically, this situation manifests in a generalised inflammatory response, which in the literature is referred to as systemic inflammatory response syndrome (SIRS). Further spread of cytokines to the sites where inflammation does not occur results in injury to the function of the endothelium, like in localised inflammatory reaction, only in multiple distant locations, in other tissues not affected by the infection or injury. Cytokines, due to their systemic action on the endothelium, change the proportions of the substances which regulate clotting and fibrinolysis produced with its participation and impact the occurrence of organ failure. By stimulating clotting and inhibiting natural fibrinolysis, they cause thrombosis in the area of microcirculation and increase organ injury [3,4,5,6]. We can then clinically diagnose such syndromes as disseminated intravascular coagulation (DIC) and multiple organ dysfunction syndrome (MODS); during episodes of DIC, both bleeding and clotting events occur concurrently, which poses considerable issues regarding therapeutic approaches given the need to try and balance these opposing events [7,8,9].

Clinical practice demonstrates that a patient with a diagnosed SIRS and symptoms of multiple organ failure should be placed in an intensive care unit, since any delay in the administration of adequate treatment usually results in death. Some researchers think that organ failure is not so much a result of generalised activation of inflammation, but of a lack of balance between proinflammatory factors and the triggered anti-inflammatory response. An anti-inflammatory overreaction is called compensatory anti-inflammatory response syndrome (CARS), which, according to this concept, in some patients is a dominating state leading to immunosuppression. A lack of balance between the factors that aggravate inflammation and the anti-inflammatory factors may result in the development of a mixed antagonistic response syndrome (MARS). MARS is an immunological dissonance, where surges of hyperactivity (SIRS) and immunosuppression (CARS) can periodically occur, which can become increasingly destructive, and if severe enough lead to MODS, septic shock, or a state of anergy [10]. Both CARS and MARS depend on the processes of activation and/or suppression of multiple mediator cascades (coagulation, complement system) and cellular cascades (cytokines, reactive oxygen species, arachidonic acid metabolites, nitric oxide) [11,12]. If the generalised inflammatory reaction is stopped, that means that immunoregulators have played their role. The inflammatory response reaction is a complex process that includes many elements of the systemic defensive reactions. The type of pathogen, massiveness of the infection or the extensiveness of the injury, and the patient’s immunological condition and genetic predisposition decide the host’s reaction, activating transformations which result in various clinical effects (elimination of infection, interrupting the infectious agent and chronic infection, and distorted immune response, that is, autoimmunisation, carcinogenesis). In response to infectious and noninfectious factors that cause SIRS, the organism has developed compensatory anti-inflammatory mechanisms which are responsible for maintaining the body’s homeostasis. They consist, among others, of production of anti-inflammatory cytokines (such as IL-4, IL-6, IL-10, IL-11, IL-13, and transforming growth factor-β (TGF-β). Moreover, a reduced expression of cytokine receptors and production of soluble receptors, e.g., soluble TNF receptor p55 (sTNFR I), soluble TNF receptor p75 (sTNFR II), soluble IL-1 receptor type 2 (sIL-1R II), and IL-18 binding protein (IL-18 BP), or the presence of receptor antagonists, e.g., IL-1 receptor antagonist (IL-1 ra), were reported [13,14,15]. Cytokines which have the ability to inhibit the synthesis of TNF-α, IL-1, and other proinflammatory cytokines are considered anti-inflammatory. The oxidative stress indicated in the pathomechanism of SIRS, which is a symptom of disorders of free radical metabolism, additionally triggers a response intended for its elimination. It is related to the occurrence within the body of all processes leading to deactivation of active forms of oxygen and is called an antioxidative barrier.

It should be emphasised that anti-inflammatory mechanisms are induced by an increased concentration of proinflammatory cytokines, which occurs in a body subjected to inflammation after antigen stimulation by factors that cause SIRS. For people researching this problem all over the world, imitating the systemic anti-inflammatory response became a premise for investigating new therapeutic strategies, undertaken in order to treat septicaemia, septic shock, or other clinical units caused by broadly understood injury. Directing these efforts to neutralising the action of bacterial toxins and endogenous inflammation mediators recently became an incentive for intensive research on the possibility of immunomodulation and immunotherapy of patients and their practical use in various diseases. It seems, however, that various schemes created to inhibit the inflammatory and immune reaction of the body do not provide the expected results. This is probably caused by the fact that cytokines and ROS are both pathogenic and protective in the pathomechanism of SIRS. Moreover, so far, all the transformations and mechanisms of the known pro- and anti-inflammatory mediators are not understood yet; new immune modulators, which could be synthesised in the body in the course of a generalised inflammatory reaction, have not been discovered yet; and our knowledge in this field remains incomplete. The goal of a clinician, thus, remains to conduct various types of therapeutic interventions on various pathogenic levels of SIRS, which may prove helpful in restoring disturbed homeostasis, and this becomes a challenge to researchers in this field.

## 2. Pathomechanism of Immune Response in SIRS

Current studies concerning the systemic immune reaction syndrome demonstrate the enormous complexity of pathophysiological mechanisms which accompany it. The generalised inflammatory reaction is related to the release of proinflammatory mediators, which are supposed to serve in the defence of the organism, but when produced in excess may harm it. That is why activating a compensatory anti-inflammatory response will determine the maintenance of homeostasis processes. In pathogenesis of SIRS, such factors as virulence of the microorganism, extensiveness of the injury, patient’s comorbidities, or polymorphism of genes controlling cytokine response and other immune response effectors are also non-negligible. The “cytokine storm” that has been recently considered to be the basis for the pathogenesis of SIRS affects macrophages, neutrophils, dendritic cells, and vascular endothelium cells, leading to their activation by cytokines created by CD4 Th1 monocytes and lymphocytes. Activation of antigen-presenting cells (APCs) through Toll-like receptors (TLRs) causes an increase in the synthesis of proinflammatory cytokines (TNF-α, IL-1, IL-6, IL-8, IL-12) [16,17,18].

The proinflammatory reactivity which then develops to maintain homeostasis in the body causes a feedback activation of anti-inflammatory mechanisms that stop it, leading to the synthesis by CD4 Th2 lymphocytes of anti-inflammatory cytokines (IL-10 and IL-13, among others) and to appearance in circulation of cytokine inhibitors (soluble receptors, receptor antagonists) [13,14]. Therefore, the CD4 helper lymphocytes may create cytokines with both a pro- and anti-inflammatory profile.

### 2.1. Immune Response to Infectious and Noninfectious Stimuli

The infectious or noninfectious factors which cause SIRS activate complex humoral and cellular mechanisms of systemic immune response (including activation of the complement, clotting and fibrinolysis, activation of neutrophil granulocytes and macrophages, and stimulation of the arachidonic acid metabolism), stimulating the production of endogenous inflammatory mediators. Among them, proinflammatory cytokines and reactive oxygen species (ROS) are considered to play key roles in injuring tissues and organs [19,20,21,22,23,24]. Cytokines are soluble glycoproteins with a low molecular weight, exerting their localised activity through paracrine or autocrine pathway (in the vicinity of the place they are secreted) or endocrine pathway (in this case, they have a systemic character). Recently, much attention has been paid to the role of ROS in tissue injury in the course of generalised infections or injuries, and in particular to disorders of the balance between their production and deactivation, which as a consequence leads to “oxidative stress”. In the situation where the body’s immune reaction is adverse and uncontrolled, cytokines and ROS released in excess disturb homeostasis and lead to many complications. We frequently observe in this situation the development of such clinical conditions as adult respiratory distress syndrome (ARDS), shock, renal failure, or multiple organ dysfunction syndrome (MODS) [25,26,27,28,29,30]. This syndrome may also develop as a complication of septicaemia, widespread multiorgan injury, severe burns, hypovolaemic shock, etc. The consequence of a severe burn injury is a burn disease, which defines local and systemic changes caused by a burn injury, with subsequent development of SIRS and accompanying metabolic disorders. The multiorgan injuries which are observed then are a consequence of localised and generalised inflammatory response of the organism and coexisting changes to the neurohormonal system. Stimulation of macrophages by damage-associated molecular patterns (DAMPs), released from the tissues injured by a burn and a whole series of complex biochemical reactions, which are a consequence of activation of these nonspecific response cells, leads to intensified production of endogenous mediators of inflammation secreted by neutrophils, fibroblasts, and vascular endothelial cells. The developing inflammation causes the movement from the bloodstream to the inflammatory focus of neutrophils, which have the task of cleaning the burn injury (they use phagocytosis to absorb pathogenic bacteria and elements of destroyed tissue). This migration of leukocytes to the wound area is stimulated by the elements of decomposition of collagen, elastin, components of the complement system, and TGF-β, TNF-α, IL-1, and platelet factor-4. Macrophages also produce cytokines (TGF-β, IL-1, insulin-like growth factor 1 (IGF-1), fibroblast growth factor 2 (FGF-2), and platelet-derived growth factor (PDGF)), which exert significant influence on the wound healing process due to the fact that they stimulate angiogenesis, migration and proliferation of fibrocytes, production of collagen, and wound shrinkage [31,32,33]. Undoubtedly, the process of wound healing depends significantly on the presence of cytokines, which control the activation of the gene responsible for the issues of cellular migration and proliferation. They take part in the wound-healing process and activation of the immune system during the acute phase response.

Endogenous mediators of the systemic inflammatory response, which occurs in the presence of an infection or without it, seem to be biochemical markers which are the closest related to MODS. Data in the literature indicate that increased concentrations of pathogen-associated molecular patterns (PAMPs), DAMPs, TNF-α, and other proinflammatory cytokines in the blood of severely ill patients are often related to an increased frequency of organ injuries [34]. Lantos et al. assessed a prognostic value of cytokines and DAMPs tested in the blood of patients with a severe burn injury, indicating that the patients who did not survive presented higher concentration of high-mobility-group box protein 1 (HMGB-1) and IL-10 at hospital admission compared to those with a favourable course of the disease [35]. Therefore, multiorgan activity of endogenous mediators of inflammation (proinflammatory cytokines and reactive oxygen species, among others) and significant displacements in the fluid spaces of the body result in a series of pathophysiological and clinical outcomes. They include, among others, a decrease in the volume of circulating blood, reduced cardiac output, hypercoagulability of blood (frequently complicated by the DIC syndrome), and progressing organ hypoperfusion with typical clinical symptoms (e.g., reduced diuresis and increasingly fluctuating consciousness). A full clinical picture of a developed burn disease as a typical case of SIRS with noninfectious aetiology is supplemented by ARDS, progressing circulatory depression and developing burn shock with the development of MODS. This syndrome may be a primary disease as the final effect of shock, but it is also possible for it to occur secondarily, when the release of endogenous mediators of inflammation (from the foci of infection and/or necrosis of the burned skin) becomes the reason for repeated development of adverse pathophysiological phenomena, which then lead to MODS. Luckily, these cases of MODS development are rarely diagnosed in severely burned patients, among others, due to modern methods of treatment (fluid resuscitation, early excision of necrotic tissue with subsequent skin graft, antibiotic therapy) and the use of currently recommended guidelines for diagnostics and treatment of sepsis, septic shock, and MODS [36].

### 2.2. Inflammatory Response to SIRS-Inducing Stimuli

Cytokines are the physiological transmitters of SIRS, and neutrophils, macrophages/monocytes, and endothelial cells are effector cells of this response. Neutrophils, as the first among the professional phagocyting cells, appear at the focus of inflammation a few minutes after the action of an inflammatory stimulus, whereas macrophages appear a few hours later and remain present at the focus of the infection for a longer time. The endogenous mediators of inflammation (including proinflammatory cytokines, reactive oxygen species, eicosanoids, and nitric oxide, among others), which are produced in excess by macrophages and neutrophils in conditions of uncontrolled generalised inflammatory response, injure the host cells. In the early period of the infection, the defensive role of the macrophages consists of phagocyting the microorganisms, in which they are supported by neutrophils. Macrophages are then activated solely by binding with a ligand present on bacteria (lipopolysaccharides) and with other ligands, similarly to neutrophils. At the further stage of SIRS, these cells are a source of cytokines, which, in septicaemia or generally understood injury under conditions of disturbed homeostasis, exert adverse effects on the organism. In complex pathogenesis of SIRS, increasing attention is being paid to the activity of proinflammatory cytokines (TNF-α, IL-1, IL-6, IL-8, IL-12, IL-18) released by activated macrophages, neutrophilic granulocytes, and dendritic cells, and ROS created in the course of “respiratory burst” of neutrophils [19,20,21,37,38]. The phagocyting, dendritic, and vascular endothelial cells, activated by bacterial antigens through TLRs, release many proinflammatory mediators, the excessive expression of which may lead to tissue injury. The released proinflammatory cytokines stimulate white blood cells, mainly macrophages, to produce significant amounts of nitric oxide, through long-term activation of inducible synthase (iNOS). Simultaneous induction of xanthine oxidase and NADPH oxidase, and the created superoxide anion radical O_2_^−^ causes mast to release cells selectin activators (histamine, thrombin), which facilitate the first contact of leukocytes with endothelial cells and the process of their rolling along the vessels. The simultaneously released adhesion activators, platelet-activating factor (PAF), leukotriene B_4_ (LTB_4_), and C5a enable the adhesion of leukocytes to endothelial cells, and then their migration to the location of inflammation [39,40]. The macrophages activated during SIRS also start to secrete TNF-α, IL-1β, IL-12, and IL-18. The last two, in turn, activate T lymphocytes and NK cells, stimulating them to produce IFN-γ, which, together with IL-12 and with the participation of dendritic cells (DC1), induces the differentiation of CD4^+^ lymphocytes towards the Th1 phenotype, while the Th1 lymphocytes start to produce IFN-γ, IL-2, and lymphotoxin α (LT α). Moreover, INF-γ inhibits the differentiation of CD4^+^ lymphocytes towards the Th2 phenotype, which could lead to synthesis of IL-4, IL-5, and IL-10 (strong inhibitors of Th1 type cytokines). IFN-γ additionally possesses the properties of activating the bactericidal activity of bacteriophages by inducing the production of TNF-α, ROS, and reactive nitrogen species (RNS) and increasing the expression of MHC antigens. In addition to acting on macrophages, the activation of neutrophils is an additional mechanism through which IFN-γ may protect the body against infection [37,39], whereas the TNF-α and IL-1 generated by activated macrophages are strong activators of neutrophil granulocytes, stimulating them, among others, to a respiratory burst and production of ROS. The described dependencies are shown in Figure 1.

### 2.3. Neutrophil Apoptosis—The Main Protective Mechanism against ROS and Proteolytic Enzymes of Activated Neutrophils

Apoptosis, which is controlled cell death, is one of the physiological processes on which the correct development and functioning of the organism depends. Alternatively, uncontrolled death of a neutrophil in the necrosis process is always related to the release of cell contents, which may potentially cause tissue injury. Apoptosis, unlike necrosis, is characterised by the fact that the cells themselves control their death in a manner that ensures that their self-destruction does not harm the remaining elements of the body. Neutrophils, due to their free radical and enzymatic potential, are special cells that must be subjected to precise mechanisms of controlled deactivation. Resting neutrophils live for a short time (on average 8 h), whereas action of activating factors (LPS, among others) will extend the life of these cells while simultaneously increasing their cidal potential. That is why examining the factors that impact the process of apoptosis of neutrophils is a significant element of the analysis of the entire inflammation process. It was demonstrated that IL-1, IL-2, IL-6, IL-8, IL-18, TNF-α, IFN-γ, IFN-α, IFN-β, GM-CSF, G-CSF, and IGF-1 are able to delay apoptosis, having an anti-inflammatory effect in the acute phase of inflammation, while of these factors, TNF-α and IL-6 have bifunctional properties, i.e., can activate neutrophil apoptosis under certain conditions [41,42]. Subsequent reports from the literature, which indicate the Treg lymphocytes’ ability to induce cell death by a mechanism that depends on the release of cytolytic granules (perforins/granzymes), allow us to postulate that Treg lymphocytes may impact the life of neutrophils [43]; however, the manner of induction of apoptosis of neutrophils is a poorly researched issue. Undoubtedly, this process mainly involves CD95 (Fas/Apo-1), TNF-R1 receptors, and DR3, DR4, and DR5 death receptors [44]. It should be remembered that biochemical, enzymatic, and morphological changes occur in cells undergoing apoptosis, which may be recorded in a precise manner (among others, by assessing the changes of integrity of cell membrane, combined with staining dead cells with propidium iodide or by analysis of changes of expression of the CD16 receptor on the surface of neutrophils, where an increased percentage of neutrophils with a low expression of this receptor constitutes evidence of an intense process of cell apoptosis).

Both those neutrophils that were not activated and those which have fulfilled their physiological role in the neutralisation of the pathogen undergo destruction through the process of apoptosis; a characteristic condensation of cytoplasm and concentration of nuclear chromatin and fragmentation of DNA can be then observed. Such additional protection of cellular membranes prevents the toxic contents of a neutrophil from being released into the extracellular space. In the sites of ongoing inflammatory processes, the neutrophils, after performing their principal activities, are removed from the “battlefield” by local macrophages; this, however, only applies to these cells which underwent an apoptosis process: their proteolytic and oxygenating potential is then effectively neutralised [45]. Therefore, a controlled, safe neutralisation of neutrophils through the process of apoptosis induction is probably the main mechanism protecting against the consequences of oxidative stress and release of proteolytic enzymes by activated neutrophils [43]. Thus, when presenting the complex pathomechanism of immunological reaction in the course of SIRS, one cannot forget about the role of T lymphocytes, and, in particular, the subpopulation of Treg CD4^+^CD25^+^ lymphocytes. This subpopulation of cells demonstrates immunomodulating properties and decides, among others, when not to respond to antigens, when a given antigen is harmless, or is an own antigen (regulation of the development of SIRS by silencing the inflammatory process). Demonstrating the possibility of modulating the activity of human neutrophils through Treg CD4^+^CD25^+^ lymphocytes would discover new, previously unknown scopes of intercellular interactions in the course of the body’s inflammatory reaction. As reported by Lewkowicz et al., the Treg CD4^+^CD25^+^ lymphocytes inhibit the production of ROS and cytokines (TNF-α, IL-6, and IL-8, among others) by neutrophils and induce their apoptosis, and lipopolysaccharide (LPS) significantly increases the strength of Treg lymphocyte action. Moreover, the induction of neutrophil apoptosis by these cells involves mechanisms that depend on both the release of cytolytic granules (perforins/granzymes) and Fas/FasL interactions [43]. It also seems that Treg lymphocytes may be partially responsible for ineffective elimination of pathogens but may also prevent the effect of destruction of own tissue related to excessive immunological activation. The results of recent studies indicate that Treg cells may infiltrate the sites where the inflammatory process occurs and limit the Th1 type answer, which is necessary to combat the infection [46]. It is considered that the molecules of PAMP pathogens and dendritic cells that have the ability to produce appropriate chemokines regulating the distribution of Treg cells in the body may have an impact on the regulation of activity of regulator lymphocytes [47].

### 2.4. Immunoregulatory Abilities of Neutrophils

Neutrophils are the first-line responders of the innate immune system, playing a key role in the destruction of invading pathogens. However, the literature also contains reports on interactions between neutrophils and lymphocytes in the course of immune response. Leukocytes also participate in humoral immunity through refined cross-communication with other immune cells [48,49,50]. Interesting results were provided by research on the impact of neutrophils on the activity of cytotoxic cells and natural cidal cells; activated neutrophils are the main cells in the early stages of tumour development. Moreover, the inhibiting action of neutrophils and negative feedback occurring between them and NK cells is pointed out, as well as the participation of neutrophils in the inhibition of antibody-dependent cellular cytotoxicity (ADCC). ADCC can be mediated by various types of immune cells, including neutrophils, the most abundant leukocyte in circulation. Neutrophils express a number of Fc receptors, including Fcγ- and Fcα-receptors, and can therefore kill tumour cells opsonised with either IgG or IgA antibodies [51,52]. Therefore, neutrophils represent intriguing and important effector cells, in particular in patients who are treated with tumour-targeted antibodies and form a promising cellular therapeutic target for the improvement of effectiveness of these antibodies. There is also significant evidence for functionally separate subsets and widespread cellular plasticity which enables a series of roles depending on cellular location and inflammation [53,54]. These immune cells may be stimulated and/or activated by many stimuli, such as proinflammatory cytokines, chemokines, growth factors, and pattern recognition receptors (PRRs)—mainly c-type lectin receptors, opsonins (C3a and IgG), and G protein-coupled receptors [55,56].

Research on TNF-α and septicaemia suggests that the synthesis of large amounts of proinflammatory cytokines during SIRS is rather harmful for the body, whereas blocking the cytokines could prevent the development of generalised infection. However, attempts to administer anticytokine therapy (anti-TNF-α) in such patients did not prove very effective [57]. Therefore, a model of pathomechanism of SIRS development was proposed that could reconcile these contradictory data. In this concept, the initially low production of cytokines in initial stages of the development of the disease, when the infection is localised, leads to ineffective immune response of the body. This results in incomplete destruction of bacteria (a process in which the Treg CD4^+^CD25^+^ lymphocytes may participate partially) and their multiplication and dissemination in the entire body; due to the significant antigenic strength of the surviving pathogens, this may lead, in the further course of the disease, to massive cytokine activation, which ends in the development of SIRS and septic shock and the patient’s death. The discussed concept therefore explains the reports of researchers who described significantly increased levels of cytokines in the blood of patients with septic shock (most clinical studies are conducted when the patients are already diagnosed with shock, that is, in the late stage of the disease; therefore, the levels of cytokines in the blood in these cases are increased and this is the result of multiplication of microorganisms and survival of bacterial antigens). The overproduction of cytokines at the beginning of the disease, however, is related to the effective answer of the host, which leads to the killing of microorganisms and their elimination, which leads to a decreased concentration of cytokines in the further course of the infection and lack of development of generalised inflammatory response. The presented concept is illustrated in Figure 2.

### 2.5. Adhesion and Migration of the Endothelial–Leukocyte Complex

Recruitment of neutrophils from the blood to the tissue affected by generalised inflammatory reaction is regulated in vivo by highly specific and selective mechanism of diagnosis. Such cytokines as TNF-α or IL-1 induce expression of receptors on the surface of neutrophils and on endothelial cells that participate in the process of adhesion of neutrophils. The inducing of receptors, such as intercellular adhesion molecules (ICAM-1, ICAM-2) on endothelial cells, integrins (e.g., complement receptor 3 (CR3 or CD11b/CD18)) on the cytoplasmatic membrane of neutrophils, and selectins (on endothelium and neutrophils) causes the neutrophils to adhere to vessel walls [58], whereas the chemotaxis of neutrophils is possible due to the activity of fMLP, C5a, PAF, LTB_4_, and IL-8. These substances bind to receptors on the surface of neutrophils, activating these cells. Chemoattractants are responsible for the accumulation of neutrophils at the site of infection and their stimulation to the secretion of lysosomal enzymes and production of reactive oxygen species. During chemotactic stimulation, free oxygen radicals (produced within the area of the cellular membrane) may penetrate from the phagolysosome to extracellular spaces and to the surrounding tissue, causing injury, which in cases of disturbed homeostasis is a manifestation of systemic inflammatory response and clinically may result in the development of ARDS and MODS. Because neutrophilic granulocytes are the main and dominating pool of phagocyting cells in blood, it is assumed that their number and functional status decide on the total level of production of ROS in the body in the course of SIRS. The neutrophils circulating in the blood may occur in three physiological states, that is, as resting cells, as activated cells, and as primed cells, where they present much stronger response to stimulating factors. This applies to most of their functions (adherence, chemotaxis, phagocytosis, intracellular degranulation, and release of ROS and cytokines, among others).

### 2.6. Neutrophil Priming

Various endo- and exogenous factors may prime neutrophils. Among endogenous factors, TNF-α is considered to be the one that exhibits the strongest priming activity; in addition, GM-CSF, IL-8, C5a, LTB_4_, and IFN-γ [59,60] are also indicated. Among exogenous stimulators, bacterial LPS, fMLP, and opsonin-coated pathogens are listed. One of the mechanisms that lead to reinforcement of respiratory burst (observed in primed granulocytes) as a result of stimulation by TNF-α or LPS is the increase in expression of flavocytochrome b_558_ in cellular membrane through exocytosis of intracellular granules in a process regulated by p38 mitogen-activated protein kinase (MAPK) [60]. The presented phenomenon of neutrophil priming may have significant practical importance in pathological conditions in vivo, since a primed granulocyte overreacts. As a result of priming of neutrophils in peripheral blood, overproduction of ROS and proinflammatory cytokines may occur, accompanied by the inhibition of apoptosis of these cells [61]. Increased generation of ROS may disturb the homeostasis of the body by overcoming the antioxidative barrier and increase the peroxidation of lipids of cellular membranes and lead to point mutations of DNA (the possibility of carcinogenesis) [19].

In primed and activated neutrophils, the synthesis of reactive oxygen compounds is also possible, such as taurine chloramine (TauCl), which, due to its longer half-life, has a wider scope of activity than ROS [62]. Oxygen radicals and the lipid oxidation products are cytotoxic and may injure the vascular endothelium. This may result in an increased aggregation of platelets, as well as disturbances of arachidonic acid metabolism of cells that take part in the inflammation. Large concentration of granulocytes in the vessel additionally intensifies oxidative stress, which may not be prevented by the presence of extracellular antioxidative enzymes—superoxide dismutase (SOD) and natural antioxidants (vitamin E). This leads, at a subsequent stage, to complete free radical oxidation of polyunsaturated fatty acids with further consequences (change of antigenic properties of proteins or inhibiting their enzymatic activity) [19].

## 3. Vascular Endothelium and Inflammation—Potential Therapeutic Options

The endothelium constitutes a barrier between blood and tissue. It constitutes a single layer of cells with a total mass of approximately 1 kg, which covers the internal surface of blood vessels, and in the area of microcirculation they remain in direct contact with cells of organs. Currently the endothelium is considered to be a very highly active endocrine organ: it produces and releases various substances that may exhibit both vasodilatory (nitric oxide, prostacyclin. bradykinin, natriuretic peptide C) and vasoconstrictive (endothelin-1, angiotensin II, thromboxane A_2_, oxygen free radicals) activity [4]. These vasoactive factors precisely regulate vascular homeostasis and enable the maintenance of an adequate blood flood through the organs and maintenance of appropriate reaction of the vascular endothelium to injury.

### 3.1. Endothelial and Local Tissue Response

The endothelium plays a very important role in inflammatory processes and in clotting and fibrinolysis. This should be remembered, in particular, in the context of the fact that the activation of inflammatory reaction is accompanied by activation of the coagulation system. It was observed that proinflammatory cytokines (TNF-α and IL-1) cause an increase in expression of tissue factor (TF) on the surface of endothelial cells and of monocytes and neutrophils, and by stimulating its release they initiate the clotting process, which leads to the creation of thrombin and fibrin deposits [63,64]. The tissue factor is a key mediator that connects the immune system with the coagulation system. According to the current concept, blood clotting is activated by contact of blood with TF, which is present in the subendothelial layer of vessel walls, on blood platelets and phagocytes. TF activates factor VII, which in turn activates factor X, which causes prothrombin to convert to thrombin, which acts on fibrinogen, converting it to fibrin. In most patients with septicaemia, fibrinolysis is inhibited, while the clotting process is continued. A key element of fibrinolysis is plasmin, created from plasminogen as a result of activity of tissue plasminogen activator (tPA). The final effect of inhibiting fibrinolysis and activation of clotting is a dynamic process of coagulopathy [65].

Such cytokines as IL-1 and TNF-α can activate MAPK/NF-κB signalling pathways within vascular endothelial cells, which leads to increased synthesis and release of procoagulant factors. Additionally, activated endothelial cells express tissue factors that trigger extrinsic coagulation pathways, resulting in a hypercoagulable state of the vascular endothelium, which can eventually lead to endothelial-related complications such as heart disease, intravascular disseminated coagulation (DIC), neurological complications, and multiple organ failure (MODS) [66,67,68]. As a result of the fact that the phagocyting cells and endogenous mediators of inflammatory reaction activated from the site of injury are released into circulation, SIRS may occur in any organ, and in particular in ones where significant contact of blood with endothelial cells may occur. Such conditions are met, e.g., by lungs, due to high blood flow, extensive microcirculation, low perfusion pressure, and rich representation of macrophages. It is, thus, unsurprising that lungs are a very frequent location of SIRS, and the degree to which they are damaged frequently decides whether the patient survives [69,70]. It should be noted that endothelial cells amplify the immune response and activate the coagulation system; they are both a target and source of inflammation and serve as a link between local and systemic immune responses [5]. Due to the possibility of synthesis of cellular adhesion molecules (CAMs), they fulfil an important role in the processes of adhesion of neutrophils to vessels in the course of SIRS. Under the influence of proinflammatory cytokines (TNF-α, IL-1, INF-γ) circulating in the blood, the adhesion molecules undergo expression both on the surface of the endothelium and on neutrophils, creating the possibility of phagocytes penetrating into the focus of infection and fulfilling the physiological role of these cells, that is, destroying the pathogen [71].

### 3.2. Nitric Oxide and Its Potential Effect on Cellular Respiration in SIRS

Nitric oxide (NO) is another endogenous mediator of inflammation that plays a role in the course of inflammatory reaction in the development of SIRS. It is synthesised in a continuous manner by endothelial cells and by leukocytes from L-arginine amino acid, molecular oxygen, and nicotinamide adenine dinucleotide phosphate (NADPH) with the participation of the nitric oxide synthase (NOS) enzyme [72]. The NO synthase occurs as a constitutive (in neurons and endothelial cells) and inducible isoform—inducible nitric oxide synthase (iNOS), which differs from the constitutive isoform by the fact that its expression is temporary. It is present mainly in macrophages, smooth muscle cells, fibroblasts, and activated endothelial cells [73]. The expression of the inducible form increases under the influence of proinflammatory cytokines, and the generation of large amounts of nitric oxide at the time is responsible for intracellular destruction of microorganisms by phagocytes [73,74]. It was demonstrated that its activity may increase in the course of septic shock, which results in the synthesis of very significant amounts of NO, which are responsible for severe, treatment-resistant hypotension [75]. The endothelial cells may also be a source of ROS in the body, whereas both endotoxemia and haemorrhage may be the release mechanism. These are oxygen radicals which have been attributed a significant role in initiating a cascade of inflammatory processes that lead to lung injury in the course of SIRS [76,77]. The duality of the action of NO depends on the location and rate of its synthesis and further directions of its transformations in the body. The anti-inflammatory properties of nitric oxide result from the activity of an enzyme—inducible NOS synthase (iNO). The expression of this form of the enzyme depends, in turn, on the mRNA transcription process, which may be regulated by a series of proinflammatory cytokines (TNF-α, IL-1, IL-8, IFN-γ), whereas anti-inflammatory cytokines (IL-4, IL-10) will inhibit this process in muscle cells, neutrophils, macrophages, eosinophils, and glial tissue cells [73]. In some conditions, excessive expression of iNOS may prove adverse for the body, which occurs frequently in septic shock. It is thought that continuous exposition of cells to a high concentration of NO has a cytotoxic effect, which leads to injury of multiple organs [78]. The proinflammatory properties of nitric oxide result from its further transformations within the body, which lead to the creation of peroxynitrite (ONOO^−^). This compound has strong oxidative properties, reacting, in particular, with thiol groups of proteins and polyunsaturated fatty acid radicals in lipids. Cytotoxicity of NO, in addition to participating in free radical transformations, also consists of blocking oxygen metabolism of the cells by inhibiting the respiratory chain and the Krebs cycle enzymes. The protonated form of peroxynitrite (HONOO—peroxynitrous acid), which is a radical that decomposes into two strongly oxidizing products (nitrogen dioxide and hydroxyl radical), is as dangerous for the organism, although some authors question the possibility of such a reaction occurring [78,79].

### 3.3. Pro- and Anticoagulatory Properties of Endothelium—Clinical Implications

The procoagulatory activity of the endothelium is its ability to synthesise such factors as endothelin-1, free oxygen radicals, tissue plasminogen activator 1, thromboxane A2, fibrinogen, and tissue factor, whereas the anticoagulatory properties depend on the synthesis of such substances as nitric oxide, prostacyclin, plasminogen activator, protein C, tissue factor inhibitor, and von Willebrand factor. The endothelium is also a place of synthesis of vascular endothelial growth factor (VEGF), responsible for pathologies of the perinatal period (e.g., retinopathy of prematurity). The released C-protein (of which the main source is the liver) becomes activated by endothelial and platelet thrombin–thrombomodulin complex in the presence of a cofactor (S-protein) [80,81]. As activated protein C (APC), it demonstrates anticlotting (reducing the production of thrombin by inactivating factors Va and VIIIa), profibrinolytic (inactivating the plasminogen activator inhibitor (PAI-1) and protecting the tissue plasminogen activator secreted by the endothelial cells), and anti-inflammatory properties. The anti-inflammatory properties manifest by inhibition of the synthesis of TNF and nuclear transcription factor NF-κB, and by blocking the adhesion of leukocytes; thanks to these properties, the endogenous APC participates in the interruption of the coagulation cycle and inflammation, which are characteristic for SIRS [82,83]. It should be emphasised that inflammation reduces the conversion of protein C in its activated form; thus, deficiencies of APCs are visible in inflammations such as septicaemia. Bernard et al. established that 85% of patients with severe septicaemia had low concentrations of protein C [84], whereas Yan et al. demonstrated that lowered levels of protein C were prognostic of lethal outcomes for patients with severe septicaemia [85]. It turned out that decreased levels of APCs in patients with generalised infection contributed to clotting and hindered fibrinolysis, which in effect ended with the occurrence of thrombosis in microcirculation, impairment of organ perfusion, and development of MODS. The aforementioned evidence has contributed to the attempts to use recombinant human activated protein C (rhAPC) (drotrecogin a) in the treatment of patients with severe septicaemia, in whom a high risk of death exists. Currently this drug is indicated for the treatment of adult patients with severe septicaemia and multiorgan distress syndrome, as an additional treatment for standard therapy. It should be noted, however, that activated protein C has failed to have any impact on survival in large-scale human clinical trials of patients with severe sepsis [86,87], whereas a controlled clinical study with the use of placebo in patients aged 0 to 17 indicated more frequent occurrence of central nervous system bleeds in patients treated with this preparation compared to the placebo group, which resulted in the study being interrupted and the use of this drug not being recommended in children below 18 years of age [88].

## 4. Immunomodulation in SIRS

### 4.1. Mesenchymal Stem/Stromal Cells—Preclinical and Clinical Cell Therapy

Vascular endothelial cells are important target cells of systemic inflammation, and their destruction is considered to be the initiating event underlying many serious complications of cytokine release syndrome (CRS). One of the scientists’ attempts to influence endothelial cells is the use of mesenchymal stem/stromal cells (MSCs); they are multipotent cells with self-renewing differentiation capacity and immunomodulatory properties, being excellent “seed cells” for cell therapy [89]. Through interaction with the host niche, MSCs are able to inhibit the immune system, promote cell survival, or induce angiogenesis, among other pleiotropic activities [90]. After transplantation, MSCs move along the blood vessels, passing through the endothelial wall, and home to the injured endothelium through their surface receptors, where they divide, proliferate, differentiate, and integrate into the damaged tissues, thus exerting their tissue repair and regeneration functions [91]. Thus, in response to injury signals, MSCs secrete a variety of mediators of tissue repair, including potent immunomodulatory, anti-inflammatory, regenerative (antifibrotic and angiogenic), and systemic homeostasis-regulating agents [92]. Notably, MSCs modulate innate and adaptive immunity and affect the functions of a wide range of effector cells, including T cells, B cells, macrophages, neutrophils, natural killer (NK) cells, and dendritic cells (DC) [93]. As an ideal therapeutic tool, human MSCs are currently used for therapeutic purposes in many clinical trials, e.g., in the cases of neurological disorders, cardiac ischemia, diabetes, and bone and cartilage diseases [94,95,96]. Due to the properties described above, MSCs are emerging as a potential therapy for intensive care unit (ICU) patients in severe condition [97]. Recent results of clinical trials demonstrated that MSC therapy is an effective strategy with long-term safety for treating patients with COVID-19 pneumonia and severe acute respiratory syndrome [98,99,100,101]. It was proposed that the use of MSCs for treating COVID-19 can ameliorate CRS by reducing the proliferation of T and B cells and maintaining the balance between proinflammatory and anti-inflammatory cytokines [102]. Thus, the immunomodulatory properties of MSCs and their derivatives (like exosomes) can improve the condition of COVID-19 patients with serious infectious symptoms caused by adaptive immune system dysfunction. Recently, studies have appeared in the literature on the cytokine profile in COVID-19 patients treated with MSCs, showing significantly reduced levels of proinflammatory cytokines (e.g., IL-6, IFN-γ, TNF-α, and IL-17A) and increased plasma anti-inflammatory cytokines levels (including IL-10 and TGF-β) observed after infusion of those stem cells. These results indicate that effective treatment with MSCs depends on achieving homeostasis in the profile of cytokines involved in the course of this clinical entity, which is an example of SIRS [103,104]. However, the Russian authors in their clinical trial reported thromboembolism as one of the most common adverse events of MSCs therapy, which requires particular caution when infusing MNCs in patients prone to thrombosis [105]. Preclinical studies demonstrate that MSC therapy can effectively repair endothelium damage and thus reduce the incidence and severity of ensuing cytokine release syndrome-induced complications [106,107,108]. Wang et al. reported that MSC transplantation can effectively suppress the activation of immune cells, reduce the bulk release of cytokines, and repair damaged tissues and organs [6]. Based on numerous preclinical studies, there is growing evidence of the potential benefits of cell-based therapies for the treatment of sepsis and ARDS; many studies are using MSCs [109,110]. Several experimental studies have indicated that MSCs may have potential therapeutic application in these clinical entities [111,112,113]. Furthermore, in the last years, a couple of clinical studies started using cell therapies for the treatment of sepsis and ARDS, and some safety and efficient results have been already published [114,115]. The mechanism of action of MSCs is that these cells release bioactive cytokines, chemokines, angiogenic factors, and/or growth factors into the extracellular media; moreover, it has been described that MSCs also release extracellular vesicles with bioactive compounds. All these factors directly secreted to the media or inside extracellular vesicles regulate intracellular pathways from different cells and can act on the innate and adaptive immune system [116]. MSCs have the ability to modulate the immune response and secrete several anti-inflammatory cytokines such as IL-4, IL-10, or IL-13 [117]. Several preclinical studies have demonstrated that MSCs are able to modify neutrophils’ behaviour, maintaining their bactericidal function but reducing injury to the host. In preclinical sepsis models, MSC therapy diminishes neutrophil infiltration into several organs such as lung, liver, gut, and kidney, reducing injury and improving organ function. Generally, the antimicrobial effect of MSCs is due to their effect on host immune cells; they have the ability to increase the phagocytic capacity of the host immune cells such as macrophages, monocytes, dendritic cells, and neutrophils [118,119]. Thus, cell therapies have shown promising results in preclinical studies. However, the heterogeneity of patients with sepsis and ARDS is enormous, and establishing a target population or the stratification of patients would help us to better determine the therapeutic effect of these therapies. Therefore, we need to await evidence that these cell therapies benefit patients with SIRS and evaluate the phase I and II results from the ongoing trials.

### 4.2. IL-6 Antagonists in Sepsis and COVID-19 Therapy

Currently, the treatment of CRS remains symptomatic (cytokine blockers and antibodies, e.g., IL-6 blockers, such as tocilizumab, sarilumab, and satralizumab; glucocorticoids; purifying blood to reduce cytokine levels in circulation) and is not aimed at reducing hyperactivation of immune cells, i.e., the primary cause of “cytokine storm”. This approach also fails to provide source cells, e.g., MSCs, which can differentiate into endothelial cells. IL-6 blockers are the most widely used cytokine blockers approved by the United States Food and Drug Administration (FDA) for the treatment of CRS. They are recombinant humanized monoclonal antibodies directed against IL-6 receptor (IL-6R), which competitively inhibit the binding of IL-6 to its receptor, thereby alleviating the symptoms of CRS [120]. Based on randomised trials of IL-6 receptor antagonists (IL6RAs) in COVID-19 patients, Hamilton et al. hypothesised that blockade of IL6R could also improve outcomes in sepsis. They conducted their genetic study on a large cohort and concluded that IL6R blockade is causally associated with reduced incidence of sepsis; similar but imprecisely estimated results supported a causal effect also on sepsis-related mortality and critical care admission with sepsis [121]. Their findings support the consideration of IL-6 inhibition in randomised controlled trials in sepsis. Wang et al. evaluated the efficacy and safety of specific interleukin (IL)-1 inhibitors, specific IL-6 inhibitors, and GM-CSF blockades in the treatment of COVID-19 patients (on the verge of sepsis) through systematic review and meta-analysis. This systematic review showed that tocilizumab, sarilumab, and anakinra (IL-1 ra) could reduce the mortality of people with COVID-19 (on the verge of sepsis), and tocilizumab did not significantly affect side and adverse effects and secondary infections [122]. In the context of the facts presented above, it should be noted that the immune responses in sepsis involve simultaneous activation and suppression of both innate and adaptive immune systems [123]. Either our understanding of immune responses is fundamentally flawed in some way or we are still a long way away from knowing when to treat and which immunomodulatory agent to use in critically ill patients with sepsis. Figure 3 will help readers visualize the complex pathogenesis of sepsis and additional potential therapeutic strategies.

Thus, diagnosis and treatment of sepsis remain significant challenges for healthcare providers globally, and gaining greater insights into key aspects of complicated proinflammatory processes that ensue during the onset and progression of disease remains a priority.

## 5. The Participation of Neutrophils and ROS in the Pathogenesis of SIRS

### 5.1. Sources of ROS and Their Regulation in Inflammation

Neutrophils circulating in the blood until the first contact with an infectious or noninfectious factor that causes SIRS are low-metabolic-activity cells. Their activation occurs in response to various stimuli (e.g., chemotactic substances, proinflammatory cytokines), which causes significant increase in oxygen consumption by granulocytes, initiating the “respiratory burst”. It is a complex cascade of biochemical reactions, the effect of which is the production of highly reactive oxygen compounds [125,126]. The release of O_2_^−^ takes place on the external surface of a cytoplasmatic membrane and occurs outside of the cell or inside of phagosomes (vesicles containing engulfed particles surrounded by a fragment of cellular membrane). Biochemical reactions which lead to the activation of NADPH of granulocytes are a complex process and include many transformations illustrated in Figure 4. The enzyme responsible for the respiratory burst in human neutrophils is the NADPH oxidase system, which catalyses one-electron reduction of an oxygen molecule to superoxide anion radical (O_2_^−^). This system is called the “electron transport chain”, that is, a system of proteins that transfer the electrons from the intracellular NADPH pool to oxygen molecules on the external surface of the cellular membrane. It includes flavocytochrome b_558_ and cytosolic proteins (p40^phox^, p47^phox^, and p67^phox^), which move into the membrane during the activation of granulocytes [125,126]. The electron donor for the reduction of oxygen is NADPH, which is created in the side pathway of the pentose cycle. NADP^+^ stimulates the conversion of glucose monophosphate in the pentose cycle; in activated granulocytes, the activation of this cycle and increased glucose metabolism has been observed. The nicotinamide adenine dinucleotide (NADH) may also serve as an electron donor for the NADPH oxidase system, but due to the much higher affinity of this enzyme towards NADPH, the latter remains, in practice, the sole cellular substrate for the enzyme [19].

The structure and location of the individual components of the NADPH oxidase system are shown in Figure 5.

### 5.2. The NADPH Oxidase System Activation

The NADPH oxidase system is capable of transferring electrons from NADPH to oxygen [125,127,128]. The main activators of PKC include intracellular Ca^2+^ and diacylglycerol, whereas for correct activity of oxidase, such low-molecular-weight proteins as rac2 (present in the cytosol) and rap1A (present in the cellular membrane) are also necessary. The activation of NADPH oxidase may be caused by such factors as TNF-α and other proinflammatory cytokines, bacterial (and also synthetic) N-formyl peptides, phorbol esters, angiotensin II, thrombin, platelet-derived growth factor (PDGF), platelet-activating factor (PAF), LTB_4_, complement component C5a, and antigen–antibody complexes [125,126,129]. The inhibitors of this enzyme include such substances as secondary lipid peroxidation products (aldehydes), IL-10, nitric oxide, nitrosothiols, and sulfhydryl groups (SH) [130,131,132]. One of the postulated systems for regulating the NADPH oxidase activity is phospholipase D, the activation of which is dependent on calcium ions and dominates in neutrophils subjected to the priming process. In turn, the effect of strengthening the respiratory burst (observed in the primed cells) may result from the activation of MAPK [133,134].

### 5.3. NADPH Oxidase-Derived ROS in Inflammation

The superoxide anion radical that is a result of reaction catalysed by NADPH oxidase in activated neutrophils is transformed into H_2_O_2_ with the participation of superoxide dismutase. This hydrogen peroxide is then used in reactions catalysed by myeloperoxidase (MPO) or metal ions for the production of highly reactive oxygen compounds (hydroxyl radical and singlet oxygen) [19,135]. Below, the reactions which illustrate the aforementioned transformations are shown:

The NADPH oxidase catalyses the transformation of oxygen into superoxide anion radical:NADPH + 2O_2_ → 2O_2_^−^ + NADP^+^ + H^+^

The superoxide anion undergoes spontaneous or enzymatic dismutation (with the participation of *SOD*) to hydrogen peroxide:2O_2_^−^ + 2H^+^ → H_2_O_2_+ O_2_

The created H_2_O_2_ is afterwards used in two systems: *MPO*-dependent and *MPO*-independent (Fenton, Haber–Weiss reactions); the hydrogen peroxide itself is not highly reactive, but in the presence of metal ions becomes very active, converting in the cell into a hydroxyl radical (OH^−^), which is created according to Fenton reaction catalysed by ferrous ions (Fe^+2^) or Haber–Weiss reaction (with the participation of Fe^+2^, Cu^+^):H_2_O_2_ + Fe^+2^ → Fe^+3^ + OH^−^ + OH (according to Fenton)
O_2_^−^ + H_2_O_2_ ^Fe+2^ _Cu+_ → ^Fe+3^ _Cu+2_ ^1^O_2_ + OH^−^ + OH (according to Haber–Weiss)

The MPO/H_2_O_2_/Cl^−^ system—myeloperoxidase in the presence of hydrogen peroxide catalyses the oxidation of halides (e.g., chloride ions) to hypohalous acid (e.g., hypochlorous acid), while simultaneously decomposing H_2_O_2_:H_2_O_2_ + Cl^−^ → H_2_O + OCl^−^ (hypochlorous acid)

The hypochlorous acid reacts with hydrogen peroxide, which leads to the creation of singlet oxygen (^1^O_2_):OCl^−^ + H_2_O_2_ → H_2_O + Cl^−^ + ^1^O_2_ (singlet oxygen)

As a very reactive molecule, the hypochlorous acid also reacts with amino groups, as a result of which chloramine is created (NH_2_Cl):HOCl + R-NH_2_^+^ → R-NHCl + H_2_O

The compounds created in the abovementioned reactions (superoxide anion, hydrogen peroxide, hydroxyl radicals, singlet oxygen, and hypochlorous acid) are commonly called reactive oxygen species Their characteristic feature is having one or more unpaired electrons, which makes them very reactive [136].

### 5.4. Neutrophil Respiratory Burst and Secondary Oxygen Transmitters

In humans, the enzyme responsible for the neutrophil respiratory burst phenomenon is the NADPH oxidase system. The physiological role of the respiratory burst is clearly defined and results in the killing of pathogens. Neutrophil extracellular traps (NETs) constitute networks of extracellular fibres mainly composed of DNA and granular proteins produced by neutrophils to entrap microorganisms in order to limit the spread of infection; NETs operate a bactericidal function thanks to the joint action of proteins—such as lysozyme, proteases, defensins, or histones—attached to their surface that disrupt the membrane permeability of bacterial cells, thus leading to pathogen destruction [137]. The mechanism of bactericidal effect of the burst was, however, not entirely explained. The superoxide anion radical and hydrogen peroxide are not reactive enough to be responsible for the destructive effect of the granulocytes in the course of this phenomenon. The participation of hydroxyl radical (OH^−^) which is created in the Fenton reaction was postulated, but most researchers are sceptic concerning the hypothesis of the significant role of OH^−^ in the bactericidal effect of the respiratory burst of neutrophils [138,139]. Activated phagocytes, in addition to O_2_^−^ and H_2_O_2_, also secrete NO, the high concentrations of which exhibit bactericidal activity. It was hypothesised that the specific cidal agent could be not the NO, but a product of its reaction with O_2_^−^ in the form of peroxynitrite [78]. Moreover, it should be noted that myeloperoxidase, present in large amounts in neutrophilic granulocytes, may oxidate chloride ions to hypochlorite, which at a further stage reacts with amines present in phagocytes (taurine, spermine) leading to the formation of appropriate *N*-chloramines (e.g., TauCL), which are more stable and also exhibit bactericidal properties [62,140]. The aforementioned secondary oxygen transmitters, due to their strong oxidative activity, in addition to cidal activity may also cause DNA defragmentation and oxidation of lipids [141]. Research from recent years has indicated that phagocytes are not the only source of ROS, released in large amounts out of the cell. There are reports which point to such cells as fibroblasts, endothelial cells, mesangial cells of the kidney, and alveolar type II cells of the lungs, which possess membrane NADPH oxidase and may produce O_2_^−^ [142]. Reactive oxygen species, due to their enormous chemical reactivity, enter very rapidly into various types of reactions. The effects of their activity in living organisms include oxidation of polyunsaturated fatty acids, damage to the structure of nucleic acids, inactivation of enzymes, or production of chemotactic factors. The largest changes concern the lipid part of the cell membrane, where peroxidation of lipids may occur as a nonenzymatic process under the impact of Fe^2+^, or as an enzymatic process, which is catalysed by lipoxygenase or cyclooxygenase [19]. In order to commence peroxidation of lipids, a small amount of ROS is necessary, since this process has a self-sustaining character. The final products of this phenomenon, in particular aldehydes, are less reactive than free radicals, which allows them to diffuse over significant distances and act as “secondary transmitters” for cellular injury by ROS [143]. Aldehydes react mainly with thiol and amino groups of proteins, with amino groups of lipids, amino sugars, and nitrogen bases of nucleic acids. Researchers have indicated that they may change antigenic properties of proteins and inhibit their enzymatic activity [144,145]. Moreover, the possibility that aldehydes may break DNA strands and that they are cytotoxic, mutagenic, and cancerogenic have all been reported [146,147] (Figure 6).

The participation of ROS in the pathomechanism of SIRS in patients with clinical symptoms of ARDS or MODS is also not well documented. This is caused by difficulties in reconstructing the presence and effects of activity of free oxygen radicals in vivo, since ROS distinguish themselves with a very short half-life and are rapidly inactivated by omnipresent antioxidative systems [147,148,149]. It is known that the presence of neutrophils in the focus of inflammation is related to the releasing of multiple active substances, in addition to ROS, such as proteolytic enzymes (elastase), vasoactive compounds (PAF, leukotrienes, PGE_2_), or substances present in a wound (M-CSF, GM-CSF) [150]. It is thought that the local concentrations of these compounds are much higher than in plasma, which is why their plasma concentrations may not reflect the phenomena which take place in the focus of inflammation. Tissue injuries observed in conditions of systemic inflammatory reaction of the body are related, among others, to the creation and action of further products of free radical reactions, such as taurine chloramine (TauCL) or peroxynitrite (ONOO^−^) [62,141]. That is why the toxic effects of these transformations may be limited, decreasing the concentration of O_2_^−^ through the use of enzyme which decomposes the superoxide anion radical, i.e., SOD, or drugs which inhibit xanthine oxidase, e.g., allopurinol. Such a strategy may be more advantageous than the use of endothelial nitric oxide synthase (eNOS) inhibitors. This is because nitric oxide demonstrates the possibility of direct inhibition of the superoxide anion radical by neutrophils by directly affecting the enzyme which generates O_2_^−^, the NADPH oxidase [151].

### 5.5. XO-Derived ROS in Inflammation

Reactive oxygen species, released at the site where large numbers of activated phagocytes gather, injure the surrounding tissue and subject them to strong oxidative stress. Pathogenic oxidative stress may be caused in the body by inflammation and tissue ischaemia (with accompanying anoxia), after which return of blood supply (with a subsequent oxygenation) may be observed, that is, reperfusion after ischaemia [152]. This last phenomenon is not a rare situation and may occur in various clinical situations to a larger or smaller extent. It constitutes one of the main complications of all forms of shock, and, by intensifying injuries of tissue and organs, may lead to death (clinically, a diagnosis of ARDS or MODS is frequently made in this situation). The role played by ROS in the pathomechanism of ischaemia-reperfusion injury is multistage. The dehydrogenase–xanthine oxidase (DH/XO) system is widespread in the body (e.g., its presence has been established in intestinal villi, liver, and lungs). In healthy tissue, dehydrogenase dominates, whereas in anoxic tissue, it is converted into xanthine oxidase, which uses molecular oxygen and forms superoxide anion radical; this reaction is, thus, the main source of ROS after reperfusion [153]. These transformations are presented in Figure 7.

### 5.6. ROS Activity in Inflammation and Neutrophil Apoptosis Regulation

The production of excessive amounts of active forms of oxygen, which occurs in SIRS after overcoming antioxidative barriers of blood and tissues, intensifies inflammatory processes through deactivation of protease inhibitors by oxygenation of SH groups of these proteins. ROS may, moreover, induce synthesis of various biologically active factors, such as PAF of endothelial cell growth factor, and also induce apoptosis of endothelial cells and cause the increase in nitric oxide synthetase activity. As a result of action of secondary transmitters, ROS function significantly as bactericidal factors, but are also strong stimulators of proinflammatory cytokines production, as a result of an increase in synthesis of nuclear transcription factors for these endogenous inflammation mediators [19,154,155]. Taurin chloramine (TauCL) acting as a secondary transmitter of reactive oxygen metabolites selectively modulates the ability of murine dendritic cells to induce the release of IL-2 and IL-10 from T lymphocytes; therefore, the mediators released from neutrophils, such as TauCL, at the site of inflammation may influence the activity of dendritic cells and macrophages, playing a significant role in maintaining balance between inflammatory processes and induction of antigen-specific response of the immune system [156]. It should be kept in mind that both nonactivated neutrophils and those which have fulfilled their physiological role in the neutralisation of microorganisms are destroyed in the process of apoptosis. Therefore, it is controlled neutralisation of neutrophilic granulocytes by inducing their apoptosis that is probably a primary mechanism that protects the body against excess of free oxygen radicals and proteases released by activated neutrophils [43,157]. The increase in intracellular concentration of Ca^2+^ causes the activation of multiple enzymes (proteases, phospholipases, endonucleases), which as a result causes disorders of the cytoskeleton and functions of mitochondria, leading to cell death. However, unlike other immune system cells (e.g., lymphocytes), an increase in intracellular Ca^2+^ concentration in granulocytes leads to inhibition of the apoptosis process [157,158]. It may be assumed that repeated stimulation of neutrophils at the focus of inflammation and an accompanying increase in calcium ions lead to increased duration of cell survival and to inhibition of the apoptosis process. Therefore, low production of ROS in the first phase of SIRS, as a result of overproduction of proinflammatory cytokines and exhaustion of functional reserves of neutrophils, with the engagement of subsequent pool of granulocytes with prolonged survival and inhibited apoptosis and activation of compensatory anti-inflammatory mechanisms, may result in the restoration of the free radical potential of these cells and, as a result, in an increase in the oxygen metabolism of neutrophils in the final stage of the disease [8] (Figure 8).

## 6. Biomarkers in SIRS—Diagnostic and Prognostic Issues

Neutrophils, as cells of innate immune response, are capable of synthesising various types of cytokines, including proinflammatory (e.g., IL-1a, IL-1b, IL-6, IL-7, IL-18, MIF), anti-inflammatory (e.g., IL-1ra, TGF-b1, TGF-b2), immunoregulatory (e.g., IFN-β, IL-12, IL-21, IL-23, IL-27), colony-stimulating factors (e.g., G-CSF, GM-CSF, SCF), angiogenic and fibrogenic factors (e.g., VEGF, FGF2, TGF-a, HGF), CC chemokines and CXC chemokines, and TNF-superfamily members (e.g., TNF-α, FasL, TRAIL, APRIL, RANKL) [159,160]. Therefore, these cells, in addition to the ability to destroy pathogens, also demonstrate immunomodulatory and repair properties [33,161]. However, the molecular mechanisms which control the expression of cytokines in human neutrophils have not been yet fully explained. Although the understanding of the method by which neutrophils influence or modulate tissue injury and repair has only started evolving recently, their participation (beneficial or harmful) after local activation in tissue remains unclear. Additionally, precisely adjusted mechanisms that regulate the recruitment of neutrophils become dysregulated during SIRS in the form of sepsis or as a consequence of broadly understood injury. Although recruitment of neutrophils is of key importance to a host’s defence, excessive representation of neutrophils may potentially result in injury to tissues in which inflammatory processes may not be occurring. Frequently, the complete elimination of neutrophils is not possible, but potential clinical benefits may be obtained by regulating or modifying their response [162].

### 6.1. Neutrophil Extracellular Traps

It should be kept in mind that after detection of bacteria in the blood, neutrophils may release their DNA in a configuration similar to a mesh, forming neutrophil extracellular traps (NETs), thus increasing the capturing capacity of the organ in which the NETs are released. They are covered by neutrophil proteases (e.g., elastase), antimicrobial molecules (e.g., histones), and also by other toxic molecules that kill pathogens; therefore, absorbing and killing pathogens seems to be the primary function of NETs [161,163,164]. However, numerous studies have also revealed the detrimental role of NETs in sepsis [165,166]. It was demonstrated that the formation of NETs starts from the production of oxidizing agents by neutrophils, which leads to the degradation of nuclear envelope and release of DNA into the cell; initially, it was postulated that neutrophils undergo lysis and die off after formation of NETs [167], although the reports of other authors indicate that the release of DNA may occur through vesicular transport and degranulation [168]. In humans, an increased number of NETs in plasma has correlated with increased lung damage and mortality, whereas lowered level of deoxyribonuclease (DNase) in plasma has led to the development of ARDS caused by sepsis [169]. It was described that the release of NETs during NETosis has an adverse impact on various human diseases, including the diseases of the respiratory, circulatory, nervous, and musculoskeletal systems, kidneys, liver, cancer, and autoimmunisation [170,171,172] (Figure 9).

### 6.2. Cell-Free DNA

Patients after severe organ injury are at risk of post-injury immunosuppression, the consequence of which may be the development of sepsis [173]. Therefore, for the prognosis of life-threatening complications and assessment of severity of these patients, the reliable biomarkers of SIRS have to be used. Among them, cell-free DNA (cfDNA) in the blood has recently gained increasing interest; elevated concentrations of this biomarker were found in serum and under physiological processes such as pregnancy and physical exercise, among others [174,175,176]. Increased levels of cfDNA have also been observed in pathological processes such as infections and sepsis [177] or thermal injuries [178], as well as trauma [179]. As a DAMP, cfDNA is increased after traumatic injuries and plays a major role in the pathophysiology of SIRS, generating various types of clinical complications [178,180]. Although the exact mechanism of cfDNA release from cells is still unclear, apoptosis, necrosis, suicidal, and vital NETosis with consecutive release of neutrophil extracellular traps (NETs) are considered as potential sources [176,179,181,182]. Trulson et al. showed, in their prospective study, that cfDNA levels in serum and plasma are highly elevated in trauma and are strongly associated with injury severity and poor prognosis of patients with multiple trauma [183]. Circulating free DNA/NETs seems to be a valuable additional marker for the calculation of injury severity and/or prediction of inflammatory second hit on intensive care units (ICUs) [184]. Finally, growing evidence supports the hypothesis that DAMPs, including high-mobility group box 1 protein (HMGB1), cell-free DNA (cfDNA), and histones, as well as neutrophil extracellular traps (NETs), may directly or indirectly contribute significantly to the development of MODS [124].

### 6.3. DAMPs

Another example of a DAMP with obvious clinical implications is HMGB-1. In the literature, increased levels of HMGB-1 have been demonstrated in children with MODS [185]. Similarly, increased concentrations of these markers were present in the blood of adults suffering from sepsis and multiorgan failure. However, the concentrations of HMGB-1 found in septic patients did not differ when compared in the groups of patients who survived and patients in whom the diseases had ended with death [185,186]. Currently, HMGB-1 is considered to be an important mediator of sepsis and potential therapeutic target in cases of MODS [34]. Both DAMPs and PAMPs activating the immune cells by TLRs lead, in consequence, to production of ROS, which promote damage to the endothelium. Cytokines and released ROS (the production of which is induced by hypoxia) lead to mitochondrial dysfunction with subsequent development of cellular disfunction and organ failure [187].

### 6.4. TREMs

Triggering receptors expressed on myeloid cells (TREMs) are a group of receptors expressed on myeloid cells (e.g., monocytes, macrophages, neutrophils). They are mainly involved in the regulation of inflammation and play an important role in the innate and adaptive immune response; activated phagocytes release the receptor that can be found as a soluble form in plasma [188]. In the literature, increased levels of sTREM-1 in septic patients have been reported, indicating that it is a reliable biomarker, the levels of which could predict survival rates in sepsis better than PCT or CRP [189,190,191]. German researchers demonstrated, in their small prospective study, that a combination of normalized IL1β plasma levels, responses to endotoxin, and soluble TREM-1 plasma concentrations at the end of surgery are predictive markers of SIRS development and could act as an indicator for starting early therapeutic interventions [192].

### 6.5. NGAL

Lipocalin-2 (LCN-2) is a glycoprotein also known in the literature under the name of siderocalin or neutrophil gelatinase-associated lipocalin (NGAL), which is secreted from the inflammatory cells and tissues as a result of neutrophil activation. NGAL has bacteriostatic properties, which play an important role in the destruction of iron during antibacterial innate immune response. In addition to its important role in the innate response, this property provides a protective role in case of injuries, systemic inflammations, and various other types of cellular stress; there are attempts to use this inflammatory biomarker in kidney and liver disorders, tumours, and inflammatory diseases of the colon [193,194]. Moreover, Chang et al. investigated the predictive value of plasma NGAL in patients with severe sepsis; according to the results of the study, this biomarker discriminated 28-day survivors from nonsurvivors on day 2 and 7 and was a relatively robust predictor of 28-day mortality prediction [195]. Chinese authors in experimental studies on animals showed that animals with septic acute kidney injury (AKI) have higher serum NGAL compared with animals with nonseptic AKI; monitoring the activities of TNF-α, NGAL, and IL-6 would make great contributions in discovering sepsis and evaluating the severity of sepsis [196]. In turn, Paul et al., in their prospective cohort study on patients with acute febrile episodes fulfilling the SIRS criteria, revealed that the NGAL sepsis screening tool with a score of >7 can be used in the emergency department to identify bacterial sepsis [197].

### 6.6. Immunohistochemistry Evaluation of Biomarkers

It is extremely important for the clinician to use reliable and available biomarkers that enable rapid diagnosis with subsequent treatment of sepsis. When fatal sepsis is suspected, autopsy and routine histology results are not very specific tools and remain unconvincing. Italian researchers, based on a review of the literature, reported that the use of immunohistochemical techniques may prove helpful and could be used in the diagnosis of postmortem sepsis, concluding that each of the studied biomarkers could prove useful in confirming or ruling out a diagnosis of sepsis in the postmortem examination, especially in the forensic setting [198]. An imbalance in the release of pro- and anti-inflammatory cytokines in response to an infectious stimulus underlies the pathophysiology of sepsis development. An intensified anti-inflammatory response of the organism may consequently lead to immunosuppression, which is particularly dangerous if it complicates any severe post-traumatic conditions. Endogenous inflammatory mediators released at that time, including chemokines and cytokines, cause the activation of the vascular endothelium with all histopathological consequences and the development of clinical entities, i.e., ARDS, DIC, or MODS. In their study, La Russa et al. reported that among a large group of recognized markers of endothelial damage from the group of adhesins, ICAM-1 (CD54), E-selectin (CD62E), and VE-cadherin are useful postmortem markers of sepsis; these authors also indicated the possibility of testing other markers for this purpose, such as angiotensin-I converting enzyme (ACE), TNF-α, PCT, VEGF, some antigens expressed on leukocyte surfaces (very late antigen-4 (VLA-4), i.e., CD49d/CD29), and enzymes contained in neutrophils granules (lysozyme (LZ), lactoferrin (LF), and s-TREM-1) [198].

### 6.7. MicroRNA

MicroRNAs (miRNAs) may play a role as prognostic and diagnostic markers in the early detection of septic patients or in the differentiation between sepsis and other inflammatory diseases. MiRNAs that can be detected in the blood represent the greatest arsenal for use as biomarkers due to their relatively easy identification and testing. However, the use of miRNAs as diagnostic and prognostic biomarkers in clinical practice is currently limited due to the low sensitivity and specificity of the methods used for their identification, with the exception of real-time quantitative PCR (RTq-PCR), which in turn is costly and time-consuming [199]. MicroRNA is still an unexplored field of knowledge, and a standardized method for identifying and measuring miRNAs has not yet been developed. This issue remains a challenge for researchers worldwide studying SIRS biomarkers for use in clinical practice. Despite these limitations, research on miRNAs continues with the imminent hope of their use in the clinic. It should be recalled that most biomarkers used for sepsis diagnosis are inflammatory biomarkers, the levels of which can be altered by other conditions such as trauma, surgery, or cancer. That is why the assessment of biomarker levels should be individualized in every clinical case. There are numerous research groups that have demonstrated an altered transcriptional expression of these small noncoding RNAs in the course of sepsis [200,201,202,203,204]. For example, miR-122 plays a crucial role in the septic process and has a higher diagnostic value than CRP and leukocyte count; it has also been shown to be a prognostic marker for sepsis, albeit with low specificity and sensitivity [203]. Guo et al. found that miR-495 was downregulated in blood samples from septic patients; the decrease was even more pronounced in patients who developed septic shock [200]. On the other hand, Sun et al. found a positive correlation between miR-328 serum levels and sepsis in human patients [201]. Zhang et al. even found that the serum level of miR-29c-3p was significantly increased in sepsis patients [202]. Han et al. evaluated the prognostic value of miR-155, finding that it could also be used for predicting the mortality and treatment outcome of sepsis-induced lung injury [204]. Thus, it was observed that certain miRNAs can be used as diagnostic or predictive markers for subsequent clinical outcome.

Up until now, no anti-inflammatory therapy has been effective in sepsis clinical trials, and little is known about the role of miRNAs in regulating neutrophil function. However, an in-depth understanding of the miRNA function within neutrophils will help to identify potential clinical applications of miRNAs as therapeutic agents. Numerous studies are being conducted around the world to answer the question of whether miRNA levels could alter inflammation. For example, Chen et al. reported in their research that miR-let-7b could regulate immunosuppression by targeting the neutrophilic TLR4/NF-κB signal during CLP-induced sepsis, which reveals novel mechanisms of the involvement of miR-let-7b in neutrophilic inflammatory activity and provides valuable therapeutic targets for severe inflammation-driven diseases, including sepsis and the current COVID-19 [205].

### 6.8. Assessment of Cytokine Concentrations

Shelhamer et al. have reported increased levels of IL-6 and IL-8 in ICU patients with a severe burn injury, which were an indicator prognostic of death [206]. There are also studies about the increase in IL-18 in burn patients, which appeared within 48 h after a burn [38]. Finnerty et al. observed that the profile of cytokines, which is characterised by increased concentrations of IL-6 and IL-12 and decreased TNF-α, in children with severe burns (>40% TBSA) was prognostic of increased risk of death due to sepsis [207]. Hur et al., assessing the concentrations of cytokines in patients with a burn injury, indicated that high (and increasing on the first day of the disease) blood levels of IL-6, antagonist of IL-1 receptor (IL-1RA), and monocyte chemotactic factor (MCP-1) are clinically useful prognostic markers of death [208]. Our own prospective research in a group of children with burns demonstrated a significant decrease in concentrations of assayed cytokine inhibitors (sTNFR I, sTNFR II, IL-1 ra) and anti-inflammatory cytokine (IL-10) following the completion of treatment, compared to their initially high levels observed at 6–24 h after the injury [209]. Despite multiple studies assessing, in clinical practice, the levels of cytokines circulating in the blood from the aspect of diagnostics of post-injury infections, the use of these markers as singular indicators to identify sepsis is insufficient due to their low specificity and sensitivity. Gouel-Cheron et al., in a prospective study on 100 patients after severe injury, reported that a combination of the assessment of clinical and immune condition of the patient with measurement of circulating IL-6 improves both specificity and positive predictive value of the used marker [210].

### 6.9. Evaluation of ROS Generation

Even though adaptive mechanisms that the organism produces in response to the factor that initiates SIRS have been well studied, there is still an ongoing search for new immunological markers which would be useful in further prognoses for burn patients [211]. The part played by ROS in the pathomechanism of SIRS in patients after a burn injury is still not well documented. Isolated reports in the literature on this subject are fragmentary and varied. Dobke et al. indicated decreased activity of the NADPH:O_2_ oxidoreductase system in patients with thermal injury [212]. Similarly, Rosenthal et al. reported a decreased respiratory burst of neutrophilic granulocytes when studying the cytosol components (p47-phox and p67-phox) of this enzyme in burn patients [213]. Other authors have observed increased oxygen metabolism of granulocytes in an adequate group of patients [214]. According to reports from the literature, an impairment of chemotaxis, adherence, phagocytosis, and oxygen metabolism and intracellular destruction of microorganism by neutrophils occur in patients with a burn injury [212], whereas in assessing the activation of neutrophils in our prospective studies, we have demonstrated similarly symptomatically higher ability of granulocytes to initiate a respiratory burst after treatment compared to the initially low production of ROS right after a burn. Moreover, our research demonstrated that children in whom, in the course of a burn disease, a hypovolaemic shock developed demonstrated significantly lower initial production of ROS compared to the children in whom shock did not occur. A one-time early analysis (at 6–24 h after the burn) established particularly high concentrations of IL-1 receptor antagonist (IL-1 ra) in patients who developed hypovolaemic shock in the first period of the burn disease [211]. Therefore, monitoring of IL-1 ra in parallel with the intensity of neutrophil respiratory burst indicates a prognostic value of the examined markers in the development of SIRS complications in children with severe burns. Additionally, we demonstrated, in a subsequent study, that the assessment of neutrophil respiratory burst using the BURSTTEST might be considered a reliable marker for differentiating the aetiology of SIRS [215].

For a physician, a very important issue is the possibility of prognosing future course of SIRS and occurrence of MODS in their patients (as a consequence of systemic infection or severe injury, among others), and also monitoring the clinical course of SIRS by multiple repeated assessment of concentrations of adequate biomarkers of inflammation. It should be emphasised that the assaying of concentrations may prove finally insufficient, which is why they always have to be considered in combination with the clinical data obtained from an interview and physical examination. For this purpose, the clinician should use various available research techniques (ELISA, chemiluminescence, flow cytometry), which prove very useful in the assessment of the aforementioned SIRS biomarkers [216,217]. Taking into account the imperfections of the laboratory markers of inflammatory reaction that have been used so far, there is still ongoing intensive research worldwide directed at discovering new, sensitive, and specific biochemical markers of systemic inflammatory reaction.

## 7. The Development of SIRS from COVID-19 and Clinical Implications

The terms cytokine storm, cytokine release syndrome, or hypercytokinaemia have been used to describe a variety of conditions that have diverse aetiologies and outcomes. The systemic inflammatory response to infectious stimuli involves the activation of leukocytes and other inflammatory cells, leading to hypercytokinaemia. Cytokines have a direct role in the activation of antimicrobial effector functions; at high-enough levels, these mediators can also have systemic activities, e.g., stimulating clotting and inhibiting natural fibrinolysis. A hallmark of a cytokine storm is persistent fever and nonspecific constitutional symptoms (weight loss, joint and muscle pain, fatigue, headache). Progressive widespread systemic inflammation leads to a loss of vascular tone that is manifested as a drop in blood pressure, vasodilatory shock, and progressive organ failure. Some of the clinical manifestations have been associated with specific cytokines: IL-6 and TNF-α are linked with fever and with constitutional symptoms, while capillary leak syndrome is thought to be driven by IL-2. The increased production (myelopoiesis) and mobilization of monocyte and neutrophil populations from the bone marrow is a response to many acute infections and cytokines. These populations are typically considered proinflammatory and are recruited to sites of inflammation where they can respond to PAMPs and DAMPs, in the case of “sterile” SIRS, by producing IL-1, IL-6, IL-12, and TNF-α. The sustained production of IFN-γ and TNF-α can lead to macrophage activation syndrome associated with hemophagocytic lymphohistiocytosis (HLH), which contributes to the anaemia that is characteristic of sepsis and almost all systemic infections [218]. HLH is a hyperinflammatory syndrome characterized by CRS, cytopenias (low blood cell counts), and multiorgan failure (including the liver); in adults, HLH is most commonly triggered by severe viral infections [219]. In addition to elevated serum cytokines, high concentrations of ferritin are characteristic of HLH. A retrospective study of COVID-19 patients found that elevated serum ferritin and IL-6 correlated with nonsurvivors [220]. However, when there is continuous, strong stimulation from harmful pathogens or excessive immune responses, the equilibrium between pro- and anti-inflammatory responses is disrupted. These early-phase cytokines can further promote the activation and release of cytokines and chemokines such as IL-2, IL-6, IL-8, IL-12, MIP-1α, and MIP-1β to cause a cascade-like reaction, thereby resulting in uncontrollable inflammatory responses [221].

IL-6 is one of the key factors and its level is positively correlated with the severity of COVID-19. The molecular mechanisms for CRS in COVID-19 are related to the effects of the S-protein and N-protein of the virus and its ability to trigger NF-κB activation by disabling the inhibitory component IκB. This leads to activation of immune cells and the secretion of proinflammatory cytokines such as IL-6 and TNF-α [222]. IL-6, playing a key role in the pathogenesis of COVID-19, becomes a target for therapeutic interventions (research on tocilizumab and other IL-6 antagonists is underway). Other possible targets for the treatment of CRS in this clinical entity target IL-1β and TNF-α. Recently, there have been reports of a CRS-like disease called paediatric multisystem inflammatory syndrome (MIS-C), the clinical manifestations of which may mimic Kawasaki disease. However, it should be emphasized that an HLH/CRS diagnosis is difficult at the early stages due to the nonspecific clinical signs and symptoms, which tends to result in missed and incorrect diagnoses. Thus, coronavirus infection results in monocyte, macrophage, and dendritic cell activation. IL-6 release then initiates an amplification cascade that results in cis signalling with T_H_17 differentiation, and trans signalling in many cell types, such as endothelial cells [221]. The resulting increased systemic cytokine production contributes to the pathophysiology of severe COVID-19, including hypotension, ARDS, which might be treated with IL-6 antagonists, i.e., tocilizumab, sarilumab, and siltuximab. In severe COVID-19 patients, the use of tocilizumab was shown to be significantly associated with a reduced risk of invasive mechanical ventilation and death [223]. The meta-analysis by Chinese authors demonstrated the strong association between elevated circulating cytokines and COVID-19 severity and mortality; they revealed that circulating levels of IL-2R, IL-10, IL-1β, IL-4, IL-8, IL-17, TNF-α, and particularly IL-6, were elevated in severe and nonsurviving COVID-19 patients when compared with mild patients [224].

Summing up, a severe acute respiratory syndrome coronavirus 2 (SARS-CoV-2) infection triggers the activation of immune system and excessive production of proinflammatory cytokines, leading to tissue damage, disseminated intravascular coagulation, acute respiratory distress syndrome, multiorgan failure, and death [225].

## 8. Antioxidative Treatment in SIRS

Current attempts of antioxidative treatment undertaken worldwide (which should be treated as supporting the primary treatment) imitate, in some areas, the mechanisms of anti-inflammatory reaction of the body to the factor that causes SIRS, which is a burn. We would like to remind the reader that the antioxidative barrier constitutes an important element of these anti-inflammatory reactions and that it is then damaged. It should be, however, emphasised that, in addition to fluid resuscitation, the most important element of contemporary, modern treatment of burns is the early excision of necrotic tissues. This procedure contributes to reducing the number of complications, lowering mortality and costs of treatment. After excision of necrotic tissues, the wounds are covered by autologous split thickness skin grafts, full or mesh, and frequently by micrografts of autologous skin [226]. In a healthy body, the antioxidative barrier depends on the presence of preventive antioxidants, that is, enzymes that lower the production of ROS, scavengers of free radicals that interrupt the development of a chain reaction (e.g., albumins, bilirubin, carotenoids, and vitamins A, C, E), and reparative enzymes that remove the effects of reaction of ROS with biomolecules and repair the membranes (DNA repair enzymes and transferases, among others). In the group of preventive antioxidants, the leading role is played by superoxide dismutase, which catalyses the reaction of oxidation–reduction leading to the transformation of the superoxide anion radical (O_2_^−^) to hydrogen peroxide (H_2_O_2_) [227]. The antioxidative properties of nitric oxide should be also kept in mind. It enters into reactions with various compounds, mainly containing unpaired electrons, such as superoxide anion radical, which is why it can be treated as a scavenger of reactive forms of oxygen. The nitric oxide inhibits peroxidative injuries by acting on free radicals that are created in the lipid peroxidation processes. At the same concentrations as vitamin E, it more effectively inhibits the peroxidation of lipids; therefore, biological concentrations of NO (1–2 mM), which are locally present under conditions of inflammation, may have effective antioxidation activity in vivo [228].

The experimental research on animals indicates a significant decrease in prevalence of SIRS, sepsis, and MODS, which is related to intravenous administration of high doses of antioxidants [229,230]. Collier et al. reported that the intravenous administration of vitamin C in patients with multiorgan injury at a dose of 3 g/day and oral administration of vitamin E at a dose of 1000 IU/day decreased the intensity of systemic inflammatory reaction, mortality, and the duration of an ICU and hospital stay [231]. Other authors, examining the effects of antioxidative therapy using N-acetylcysteine (NAC) in patients with a severe burn injury, demonstrated a significant decrease in plasma levels of proinflammatory cytokines (IL-6, IL-8) and anti-inflammatory cytokines (IL-10); moreover, these patients presented low concentrations of malondialdehyde (MDA) and required the use of catecholamines more rarely [232]. Subsequent researchers also demonstrated low concentrations of MDA (as the product of lipid peroxidation) and low activity of myeloperoxidase, as well as decreasing mortality and shortening the time needed for the healing of burn wounds in patients who received treatment with N-acetylcysteine [233,234,235]. Similar results, but for supplementation with vitamin E and C, in children with a burn injury were also presented by Barbosa et al. [236]. Successive data from the literature indicate that administering antioxidative treatment in patients with a severe thermal injury (15–40% TBSA) has resulted in shortening of the wound healing time, decrease in invasiveness of bacteria in injured tissue, and lowering of mortality [237]. These authors postulated the administration as supportive treatment in burn disease of such antioxidants as vitamin E at a dose of 400 mg/day, vitamin C at a dose of 500 mg/day, zinc sulphate (75 mg/day), allopurinol—100 mg/day, melatonin—3 mg/day (at night), and N-acetylcysteine at a dose of 500 mg/day. The role of vitamin C as antioxidant will consist of alleviating injuries of the vascular endothelium and heart muscle cells caused by ROS in the course of ischaemia-reperfusion injury (clinically varied post-injury syndromes) or sepsis. By reducing the disfunction of endothelium, vitamin C may improve tissue perfusion and reduce hypoxia and organ dysfunction. Dutch authors, when assessing the intravenous administration of vitamin C in injured patients hospitalised at the ICU, demonstrated reduced oxidative activity of neutrophils in the course of systemic inflammation reaction, as well as shorter stay of these patients at the ward [238]. The increase in survival indicator was also reported after intravenous administration of high doses of vitamin C in patients with a burn injury [239]. Therefore, antioxidative therapy in patients with severe burn injury results in advantageous results, in particular decrease in intensity of inflammatory reaction and dysfunctions of microcirculation. However, the optimum dosage of antioxidants, the time period over which they are to be administered, and the potential possibilities of combining them still remain open issues.

## 9. Conclusions

Neutrophils fulfil a very important role in the protection of our organism against pathogens, due to their enormous enzymatic and free radical potential. On the other hand, neutrophils activated in the course of CRS become a source of many endogenous mediators capable of immune modulation with resulting clinical implications [159,160,161,240,241,242]. In the course of disturbances of these processes, they may become the main effector cells that cause the injury of surrounding tissue [33]. Thus, neutrophils should be perceived not only as passive phagocytes but also as cells that constitute a bridge between innate and adaptive immunity.

A cytokine release storm with a coexisting imbalance between pro- and anti-inflammatory cytokines leads to an increased activation of the immune system and the resulting clinical symptoms (ARDS, DIC, and MODS). ROS generated during the neutrophil respiratory burst play a killing role against various kinds of pathogens, but they can also participate in pathophysiological processes leading to tissue and organ damage. Thus, cytokines and ROS are both pathogenic and protective in the pathomechanism of SIRS. Hypercytokinaemia and its main pathophysiological paths have become a starting point for the search for targeted methods of immunomodulatory treatment of various clinical forms of SIRS. The described biomarkers of SIRS related to neutrophils are of clinical usefulness, especially in forecasting of such clinical entities as sepsis, burn disease, and COVID-19.

## Figures and Tables

**Figure 1 ijms-24-13469-f001:**
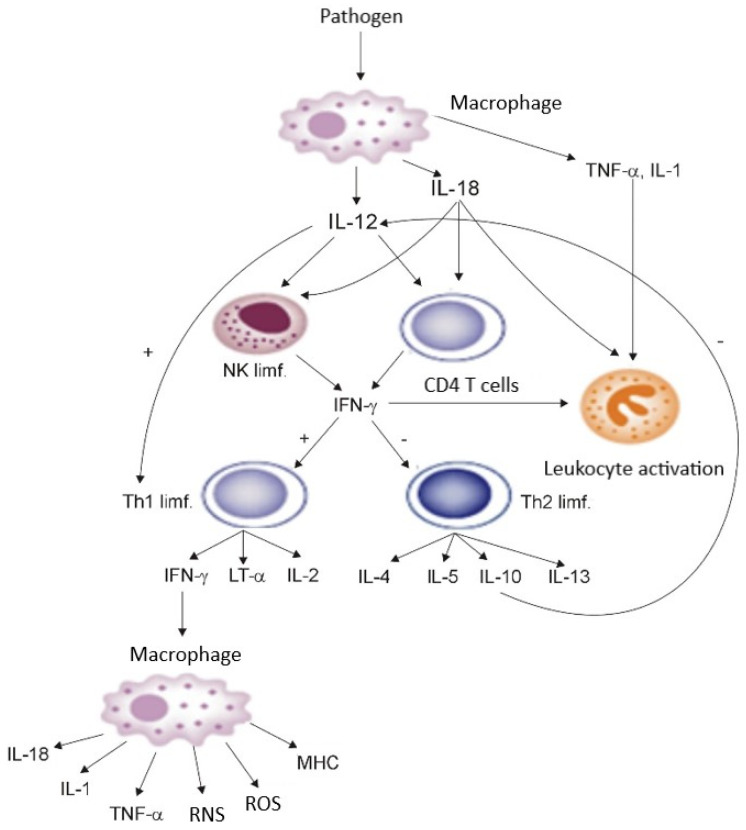
Scheme of activation of the body’s immune system by proinflammatory cytokines. Source: [37].

**Figure 2 ijms-24-13469-f002:**
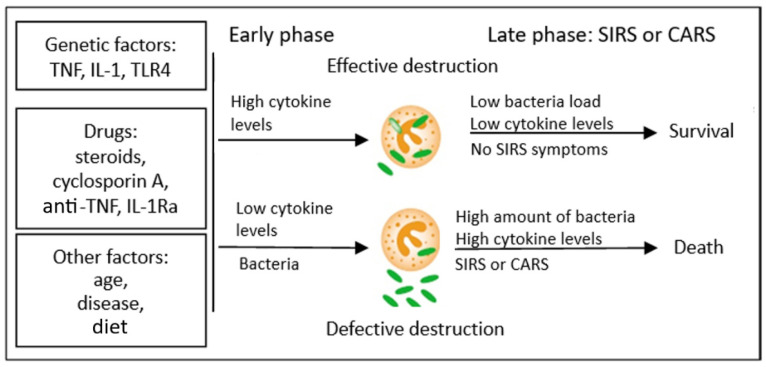
Effect of the amount of cytokine production on the development of SIRS and septic shock.

**Figure 3 ijms-24-13469-f003:**
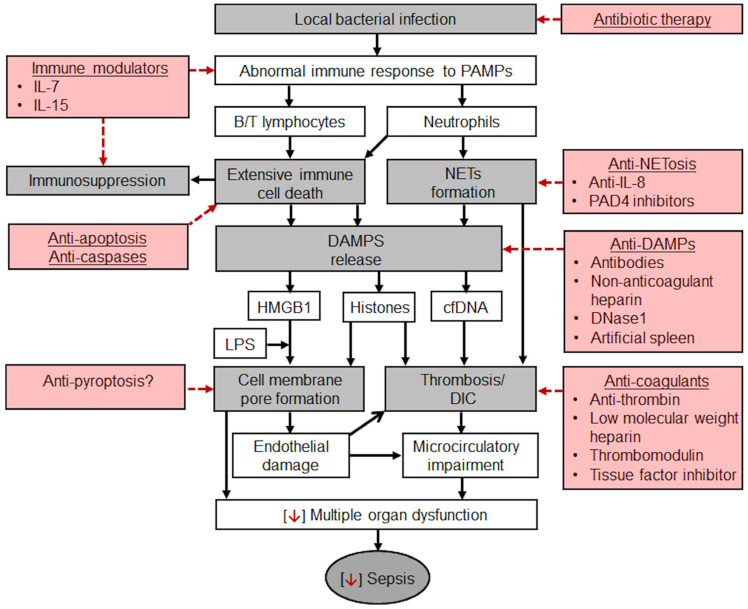
Potential pathological mechanisms of sepsis, which develops from a local bacterial infection, and potential therapeutic strategies. Gray boxes: Pathways from a local infection to sepsis. Once a local bacterial infection causes host abnormal immune responses to pathogen-associated molecular patterns (PAMPs), extensive immune cell death, including B/T lymphocytes (spleen, thymus, lymphoid tissues, and peripheral blood), and neutrophils could occur and result in immunosuppression. Neutrophils could also form NETs. NETs and immune cell death could release a large quantity of DAMPs, mainly HMGB1, cfDNA, and histones. HMGB1 can deliver LPS into cells to trigger pyroptosis by forming pores in the cell membrane. Extracellular histones could also bind to cell membranes to form pores which may cause calcium overload and subsequently endothelial damage and organ injury. Indirectly, extracellular histones activate coagulation to form thrombi in the microvascular lumen to impair microcirculation. cfDNA could serve as scaffolds for thrombosis or stabilize thrombi by increasing their resistance to fibrinolysis. Microcirculatory impairment is the major feature of sepsis and a major contributor to MODS. Red boxes: Potential therapeutic strategies. In addition to early diagnosis, prompt use of effective antibiotics, and supportive therapies, such as maintaining blood pressure and circulation, improving microcirculation, and protecting individual organs, the potential specific therapies include the combination of modulating immune status, preventing immune cell death and NETosis, and neutralizing or clearing DAMPs. These new approaches could become the leading research directions in reducing the mortality rate of sepsis. Source: [124].

**Figure 4 ijms-24-13469-f004:**
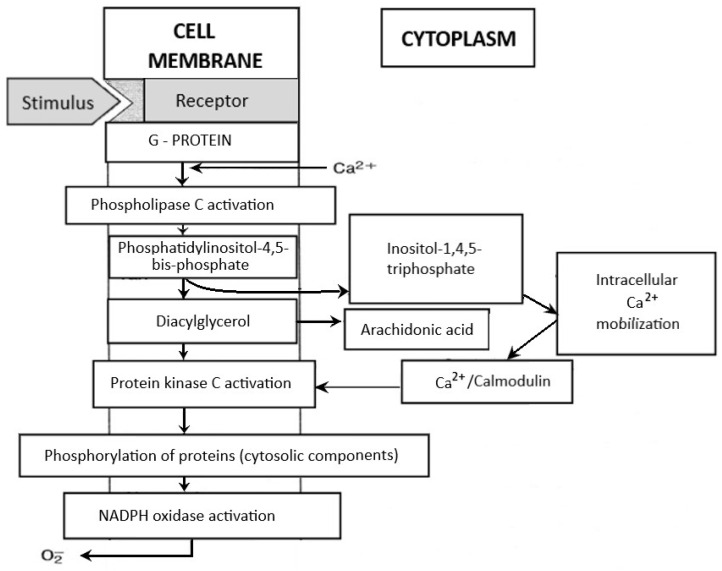
Scheme of neutrophil NADPH oxidase activation.

**Figure 5 ijms-24-13469-f005:**
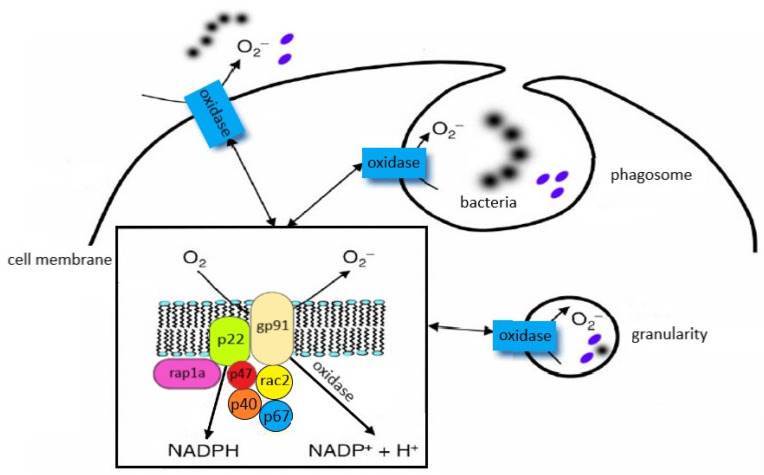
NADPH oxidase of phagocytic cells. Oxidase present in neutrophils is composed of 5 elements: p40phox (phox-phagocyte oxidase), p47phox, p67phox, p22phox, and gp91phox. In resting phagocytes, three of those (p40, p47, and p67 phox) are present in the cytosol, forming a complex. The remaining two elements (p22phox and gp91 phox) are located in the membrane of secretory vesicles, forming cytochrome b558. The separation of these two groups of components ensures the inactivity of enzymes in the cell in its resting state. After activation of the neutrophil, the p47phox unit undergoes phosphorylation with the participation of protein kinase C (PKC), and the entire complex travels to the membrane, where it combines with cytochrome b558 forming active oxidase.

**Figure 6 ijms-24-13469-f006:**
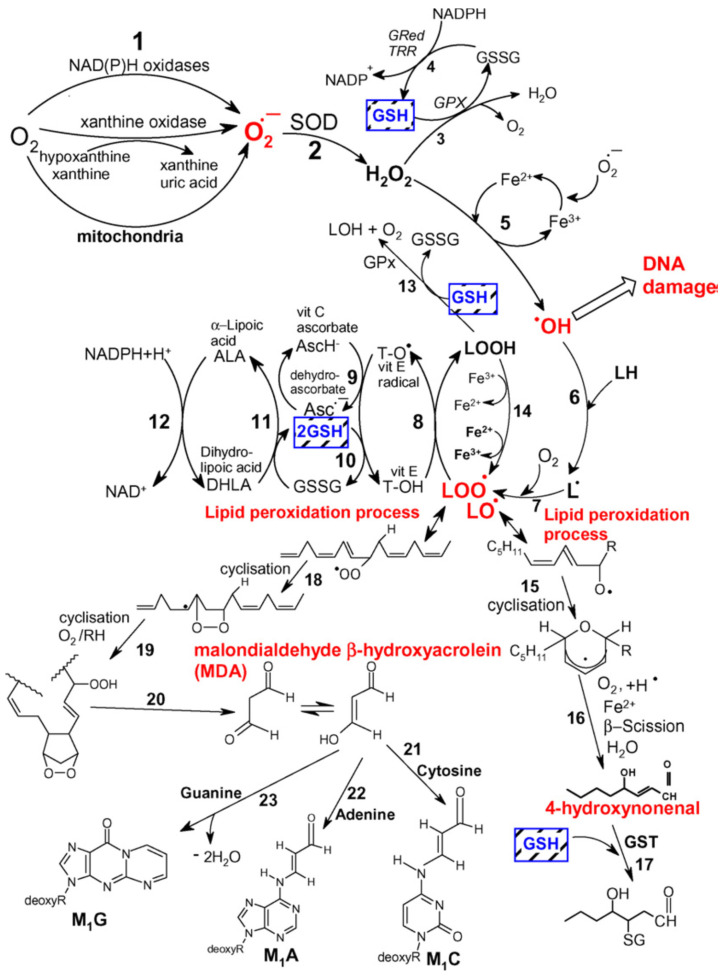
Pathways of ROS formation, the lipid peroxidation process, and the role of glutathione (GSH) and other antioxidants (vitamin E, vitamin C, lipoic acid) in the management of oxidative stress (equations are not balanced). Reaction 1: The superoxide anion radical is formed by the process of reduction of molecular oxygen mediated by NAD(P)H oxidases and xanthine oxidase or nonenzymatically by redox−reactive compounds such as the semiubiquinone compound of the mitochondrial electron transport chain. Reaction 2: Superoxide radical is dismutated by the superoxide dismutase (SOD) to hydrogen peroxide. Reaction 3: Hydrogen peroxide is most efficiently scavenged by the enzyme glutathione peroxidase (GPx) which requires GSH as the electron donor. Reaction 4: The oxidised glutathione (GSSG) is reduced back to GSH by the enzyme glutathione reductase (Gred) which uses NADPH as the electron donor. Reaction 5: Some transition metals (e.g., Fe2+, Cu+, and others) can breakdown hydrogen peroxide to the reactive hydroxyl radical (Fenton reaction). Reaction 6: The hydroxyl radical can abstract an electron from polyunsaturated fatty acid (LH) to give rise to a carbon−centred lipid radical (L•). Reaction 7: The lipid radical (L•) can further interact with molecular oxygen to give a lipid peroxyl radical (LOO•). If the resulting lipid peroxyl radical LOO• is not reduced by antioxidants, the lipid peroxidation process occurs (reactions 18–23 and 15–17). Reaction 8: The lipid peroxyl radical (LOO•) is reduced within the membrane by the reduced form of vitamin E (T-OH) resulting in the formation of a lipid hydroperoxide and a radical of vitamin E (T-O•). Reaction 9: The regeneration of vitamin E by vitamin C: the vitamin E radical (T−O•) is reduced back to vitamin E (T-OH) by ascorbic acid (the physiological form of ascorbate is ascorbate monoanion, AscH−), leaving behind the ascorbyl radical (Asc•−). Reaction 10: The regeneration of vitamin E by GSH: the oxidised vitamin E radical (T−O•) is reduced by GSH. Reaction 11: The oxidised glutathione (GSSG) and the ascorbyl radical (Asc•−) are reduced back to GSH and ascorbate monoanion, AscH−, respectively, by the ad acid (DHLA) which is itself converted to a-lipoic acid (ALA). Reaction 12: The regeneration of DHLA from ALA using NADPH. Reaction 13: Lipid hydroperoxides are reduced to alcohols and dioxygen by GPx using GSH as the electron donor. Lipid peroxidation process: Reaction 14: Lipid hydroperoxides can react fast with Fe2+ to form lipid alkoxyl radicals (LO•), or much slower with Fe3+ to form lipid peroxyl radicals (LOO•). Reaction 15: Lipid alkoxyl radical (LO•) derived, for example, from arachidonic acid undergoes cyclisation reaction to form a six−membered ring hydroperoxide. Reaction 16: Six−membered ring hydroperoxide undergoes further reactions (involving b−scission) to form 4−hydroxy−nonenal. Reaction 17: 4−hydroxynonenal is rendered into an innocuous glutathiyl adduct (GST, glutathione *S*−transferase). Reaction 18: A peroxyl radical located in the internal position of the fatty acid can react by cyclisation to produce a cyclic peroxide adjacent to a carbon−centred radical. Reaction 19: This radical can then either be reduced to form a hydroperoxide (reaction not shown) or it can undergo a second cyclisation to form a bicyclic peroxide which, after coupling to dioxygen and reduction, yields a molecule structurally analogous to the endoperoxide. Reaction 20: The formed compound is an intermediate product for the production of malondialdehyde. Reactions 21, 22, and 23: Malondialdehyde can react with DNA bases cytosine, adenine, and guanine to form adducts M1C, M1A, and M1G, respectively. Source: [19].

**Figure 7 ijms-24-13469-f007:**
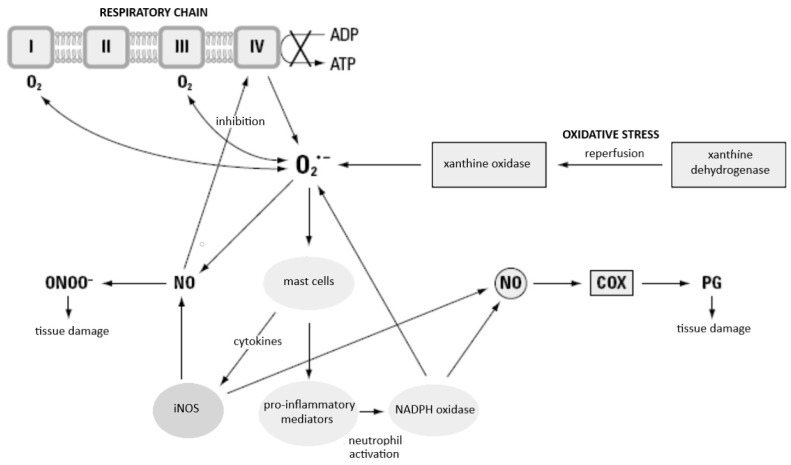
Commonly accepted mechanism of tissue damage in ischemic reperfusion syndrome (involving nitric oxide and neutrophils). Reactive oxygen species (O_2_^−^) created in the process of reperfusion cause mast cells to release cytokines, which activate iNOS. The NO formed in these conditions stops the flow of electrons in the respiratory chain and additional production of O_2_^−^ and peroxynitrite (ONOO^−^) and prostaglandins (PG), contributing to tissue injury, and the additional gathering of neutrophils and activation of NADPH oxidase intensifies this process. Source: [153].

**Figure 8 ijms-24-13469-f008:**
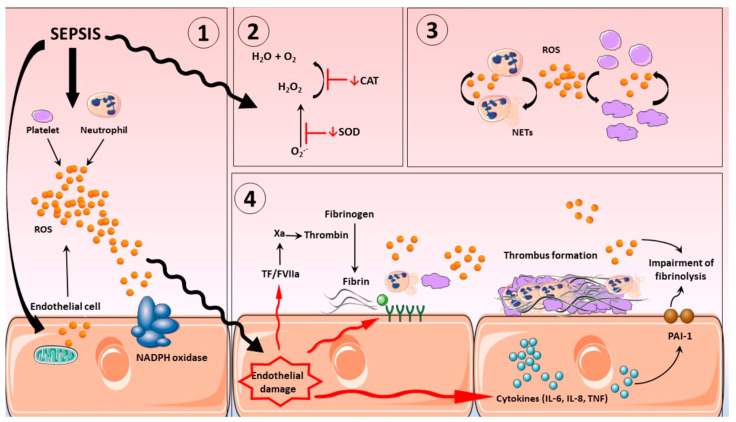
Sepsis induces oxidative stress and disseminates intravascular coagulation. ① Sepsis induces ROS release by platelets, neutrophils, and endothelial cells. The majority of excessive ROS production is generated by mitochondria and NADPH oxidase present in endothelial cells, platelet, and neutrophil. ② The overproduction of ROS results in depletion of endogenous antioxidant systems, including, but not limited to, SOD and catalase. ③ ROS release, from activated inflammatory cells such as neutrophils, and platelets further propagate inflammatory responses including further ROS production, processes that are self-sustaining and ever-expanding. ④ Damage to the vascular endothelium augments inflammatory cytokine production via ROS-mediated stress responses and activates the coagulation system and expression of adhesion molecules, all of which results in elevation of fibrin deposition, impairment of fibrinolysis, and, consequently, thrombus formation. ROS: reactive oxygen species. CAT: catalase. SOD: superoxide dismutase. TF: tissue factor. NETs: neutrophil extracellular traps. Source: [8].

**Figure 9 ijms-24-13469-f009:**
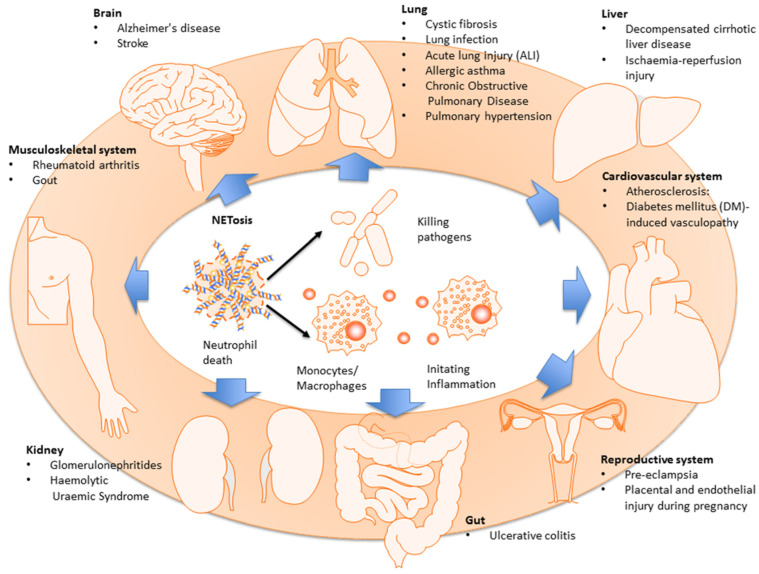
Involvement of NETosis in organ injury. Accumulating evidence now points to an important role of NETosis in infectious and noninfectious solid organ injury. Neutrophil invasion into brain parenchyma and release of neutrophil extracellular traps (NETs) have been established in the pathophysiology of Alzheimer’s disease to cause destruction to neural cells and blood–brain barrier. Abnormal NETosis activity and reactive oxygen species (ROS) response, a key element to NETosis initiation, were observed in stroke patients. The degree of neutrophil infiltration, NET formation/component (e.g., cell-free nucleosomes), and NETosis have been found to correlate with the severity of a range of lung diseases, including cystic fibrosis, acute lung injury (ALI)/acute respiratory distress syndrome (ARDS), and lung infection. NETosis was also shown to be involved in allergic asthma, chronic obstructive pulmonary disease, and pulmonary hypertension, wherein degree of NET formation correlates with disease severity. During liver ischaemia-reperfusion, Toll-like receptor-dependent NET release has been suggested to mediate liver inflammation and injury. Conversely, deficiency in NET release was reported in decompensated cirrhotic liver disease and could explain susceptibility to bacterial peritonitis infection in those patients. NET formation and NETosis have further been implicated in atherosclerosis and myocardial infarction, wherein NET was found in thrombi and infarct lesion and correlated with disease severity. In rheumatoid arthritis, enhanced NET release and NETosis are observed in synovial tissue, rheumatoid nodules, and skin, whilst proinflammatory cytokines and autoantibodies further aggravate neutrophil infiltration and NETosis. Neutrophils could also be potently activated by monosodium urate (MSU) crystals in gout joints and point to a potential role of NET/NETosis in gout pathogenesis. Moreover, neutrophil activation and NET deposition were also observed in colon mucosa of ulcerative colitis. Excessive neutrophil activation, NET formation, and NETosis could also be responsible for different pregnancy-related disorders, including pre-eclampsia, wherein NET deposition and NETosis in the intervillous space may damage maternal endothelium and impair foetal oxygen exchange. Source: [171].

## Data Availability

Not applicable.

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
