# Peer review of "Neutrophils and the Systemic Inflammatory Response Syndrome (SIRS)"

_ijms, 2023, doi:10.3390/ijms241713469_

Round 1

Reviewer 1 Report

In the article titled “Neutrophils and the systemic inflammatory response syndrome (SIRS)”, the authors have attempted to emphasize the importance of Neutrophils in systemic inflammatory reactions along with the involvement of ROS, biomarkers, prognosis and anti-oxidative treatment in SIRS. My comments are as follows:

1. The abstract seems to be quite lengthy making it really difficult to understand what the main point of the review is.

Advice: Reformulate and simplify it making it more precise and straightforward. 

2. The key arguments need to be worked out and formulated more plainly. The conclusion is short, and abrupt with no clear indications of the key contributions of this article. Special care needs to be taken for language and grammatical errors.

Advice: To improve the readability of the paper, I suggest diving sections into further subsections describing each heading clearly and precisely. 

However, I see value in the research topic and encourage the authors to revise and resubmit their manuscript.

Authors need to give special attention towards language and grammatical errors

Reviewer 2 Report

the manuscript presented explores some very interesting aspects, namely the functioning of the immune system during endogenous inflammation. The authors evaluate the role of neutrophils. The manuscript is well structured and very interesting. I advise self-authors to add articles on immunohistochemistry evaluating the expression of cytokines DOI: 10.1177/2058738419855226. Furthermore, the role of mRNAs in SIRS and how these molecules affect the activity of neutrophils should be investigated. doi: 10.3390/ijms23169354.

Finally add an explanation on the development of a SIRS from COVID. Is it the same as the bacterial one?

Authors must review the images and English style.

Round 2

Reviewer 1 Report

Thank you for addressing the previous suggestions. Overall the figure texts are still not properly aligned, different font size, unwanted lines with other hidden text can be seen popping out. The overall image quality is poor and does not meet the standard quality of the journal

Figure 1: Figure legend and source is not aligned. Presence of red grammar check line under Macrophage, Leukocyte and in lower panel unwanted line above ROS should be removed. Overall quality of figures can be improved as they appear to be blurry.

Figure2: Left middle panel anti-TNF is written as anty-TNF. Left lower panel the word diet is being hidden, so requires proper adjustment. In the same figure unwanted dash line is appearing under low cytokine level and bacteria.

Figure 4: Different font sizes have been used throughout. Many boxes cut out the important text words.

For example: Cell membrane, the membrane word is cut from below, PI(4,5) P2. What is the significance of two unending lines hanging from NADPH oxidase activation? Another arrow from Phospholipase C seems to merging with other.

Figure 5: Please check the Phagosome spelling, the letter “e” is missing in it. Also there seems to be some grammatical error as they have red underlines in it. Text is merging with inhibition symbol.

Figure 7: Different font sizes have been used, text images are blurry with red underlines and texts are also not properly aligned with the image.

Not acceptable in figures.

Round 3

Reviewer 1 Report

No further comments